# Revolutions in energy input and material cycling in Earth history and human history

Timothy M. Lenton[1], Peter-Paul Pichler[2], and Helga Weisz[2,3]

[1]Earth System Science, College of Life and Environmental Sciences, University of Exeter, Exeter, United Kingdom
[2]Potsdam Institute for Climate Impact Research, Potsdam, Germany
[3]Department of Cultural History and Theory and Department of Social Sciences, Humboldt University Berlin, Germany

*Correspondence to:* Timothy M. Lenton (T.M.Lenton@exeter.ac.uk) and Helga Weisz (weisz@pik-potsdam.de)

**Abstract.** Major revolutions in energy capture have occurred in both Earth and human history, with each transition resulting in higher energy input, altered material cycles and major consequences for the internal organization of the respective systems. In Earth history, we identify the origin of anoxygenic photosynthesis, the origin of oxygenic photosynthesis, and land colonization by eukaryotic photosynthesisers as step changes in free energy input to the biosphere. In human history we focus on the Paleolithic use of fire, the Neolithic revolution to farming, and the Industrial revolution as step changes in free energy input to human societies. In each case we try to quantify the resulting increase in energy input, and discuss the consequences for material cycling and for biological and social organization. For most of human history, energy use by humans was but a tiny fraction of the overall energy input to the biosphere, as would be expected for any heterotrophic species. However, the industrial revolution gave humans the capacity to push energy inputs towards planetary scales and by the end of the 20th century human energy use had reached a magnitude comparable to the biosphere. By distinguishing world regions and income brackets we show the unequal distribution in energy and material use among contemporary humans. Looking ahead, a prospective sustainability revolution will require scaling up new renewable and decarbonized energy technologies and the development of much more efficient material recycling systems – thus creating a more autotrophic social metabolism. Such a transition must also anticipate a level of social organization that can implement the changes in energy input and material cycling without losing the large achievements in standard of living and individual liberation associated with industrial societies.

## 1  Introduction

Human society has become a planetary force, approaching or even exceeding natural dynamics (Turner et al., 1990; Steffen et al., 2004). A great deal of work has been devoted to measuring the scale of human society with respect to the Earth system (Daly, 1973), especially after the introduction of new concepts such as the 'great acceleration' (Steffen et al., 2004), the Anthropocene (Crutzen, 2002) or 'planetary boundaries' (Rockström et al., 2009). Many studies assessing the human impact on the Earth system focus on rates of change in a multitude of parameters (Steffen et al., 2004). Others define a natural background against which the human impact should be measured, notably the Holocene epoch (Petit et al., 1999), during which the climate was unusually stable (and other environmental variables are argued to have been stable) compared to the preceding Pleistocene epoch with its characteristic glacial cycles (Rockström et al., 2009). Suggested metrics of human impact

on the Earth system include changes in land use (Ellis, 2011), bio-productive land capacity (Wackernagel and Rees, 1996), human appropriation of terrestrial net primary production (Vitousek et al., 1997; Haberl et al., 2007; Krausmann et al., 2013a) or the impact of human appropriation of free energy on the capability of the biosphere to generate free energy (Kleidon, 2012).

Here we propose an alternative approach to measure the human influence against a natural background, following pioneering work by Smil (1991), who first compared energy use in the biosphere and in human civilization (where 'biosphere' is taken here to be synonymous with the biota i.e. the sum total of all life on the planet). Our starting point is the fundamental ability of all life forms, from archaea and bacteria to human societies, to capture free energy and to use it for moving and transforming matter in order to sustain an internal order. Building on Smil's (Smil, 1991, 2008) characterization of energy use in the biosphere and human civilization, we expand the temporal dimension to consider the full history of transitions in biospheric energy capture, and we add a material cycling dimension, also partly inspired by Smil's work (Smil, 2014). In both Earth and human history major revolutions in energy capture have occurred, with each subsequent transition resulting in higher energy input, altered material cycles and major consequences for the internal organization of the respective systems.

In general, when a new biological mechanism of accessing under-utilised resources evolves, this can lead to profound environmental change – as shown by generic models capturing the co-evolution of life and its environment (Williams and Lenton, 2010). Indeed, in Earth history as new metabolic waste products were created or the production of existing waste products was scaled up, these waste products accumulated in the environment (Lenton and Watson, 2011). When step increases in free energy input to the biosphere occurred, the environmental consequences were sometimes dramatic and global – destabilizing nutrient and carbon cycles and the Earth's climate (Lenton and Watson, 2011). When past increases in free energy input to human societies occurred, the resulting waste products also disrupted the environment – initially on a local scale, but now globally. Here we compare the order of magnitude of energy use by human societies with the energy input to the entire biosphere throughout Earth and human history based on a common framework. A clear distinction to note at the outset is that the input of energy to the biosphere has thus far been dominated by autotrophs harvesting sunlight, whereas humans are heterotrophs and our current industrial consumption of fossil fuels is also essentially heterotrophic.

We consider a series of six past revolutions, three in Earth history and three in human history, each contingent on the previous one(s). In Earth history, we focus on the origins of anoxygenic photosynthesis, of oxygenic photosynthesis, and of eukaryotic photosynthesis especially the colonisation of the land by plants. In human history we consider the Paleolithic use of fire, the Neolithic revolution to farming, and the Industrial revolution. In each case we try to quantify the resulting increase in energy input to the biosphere or to human societies, and discuss the consequences for material cycling. Changes in energy input and material cycling in turn altered limiting conditions for biological and cultural evolution and we highlight some of the crucial biological and social consequences. We discuss similarities and crucial differences among the six energy revolutions, their underlying regulatory mechanisms and their impacts. For most of human history, energy use by humans was but a tiny fraction of the overall energy input to the biosphere, as would be expected for any heterotrophic species. All major increases in energy input to human societies were contingent on new technologies that shifted human energy and material use beyond the limits of their biological metabolism. We show that the capacity of humans to push energy inputs towards planetary scales only emerged

with the industrial revolution and that by the end of the 20th century human energy use reached a magnitude comparable to the biosphere.

After revolutions in Earth history, long-term sustainability and stability were only recovered when disrupted material cycles were closed again, through global biogeochemical recycling mechanisms (Lenton and Watson, 2011). Equally, for humans to have a long-term sustainable future within the Earth system will require both a shift to sustainable sources of energy and, crucially, the closure of material cycles (Weisz et al., 2015; Weisz and Schandl, 2008) – amounting to a more autotrophic social metabolism. We finish by advocating a research agenda that considers pathways towards a renewable and decarbonized energy system in its ramifications for material use and a prospective material cycle revolution.

## 2 Revolutions in Earth history

### 2.1 Anoxygenic photosynthesis

The first revolution in energy input to the biosphere was the origin of photosynthesis. The earliest life forms were probably fueled by chemical energy stored in compounds in their environment, but the supplies would have been small, except in unusual environments with concentrated volcanic/metamorphic activity such as deep sea vents near mid-ocean ridges (if plate tectonics started early on the Earth). Shortage of chemical energy on a global scale would thus have severely restricted the spread of chemolithoautotrophic life. The first truly global biosphere arose when early life began to harness the most abundant energy source on the planet – sunlight. Evidence for the photosynthetic fixation of carbon dioxide from the atmosphere is coincident with the first putative evidence for life on Earth >3.7 Ga (Ohtomo et al., 2014; Rosing, 1999), and perhaps as early as 4.1 Ga (Bell et al., 2015). It takes the form of small particles of graphite carbon, which have a likely biogenic origin, and an isotopic signature consistent with carbon-fixation by the enzyme RuBisCO.

The first photosynthesis was not the familiar kind, which uses water as an electron donor and produces oxygen as a waste product. Instead, molecular phylogenies suggest that several forms of anoxygenic photosynthesis evolved independently, early in the history of life, long before oxygenic photosynthesis (Blankenship, 2010). This makes energetic sense as there are several donor compounds from which it is easier to extract electrons than water, requiring fewer or less energetic photons and simpler photosynthetic machinery. Hydrogen gas ($H_2$) gives up its electrons the easiest and may thus have fuelled the first photosynthesis (Olson, 2006). Other potential electron donors include elemental sulphur ($S^0$) derived from sulphur dioxide ($SO_2$) gas, or ferrous iron ($Fe^{2+}$) dissolved in the ancient oceans (Canfield et al., 2006). The meagre supply of these compounds (relative to $H_2O$) limited the energy input to the early biosphere. For example, the present-day flux of $H_2$ emanating from volcanic processes can only support $\sim 0.1$ EJ yr$^{-1}$ (3 TgC yr$^{-1}$) of anoxygenic photosynthetic net primary production (NPP) (Canfield et al., 2006), over 4 orders of magnitude less than present marine biosphere (1800 EJ yr$^{-1}$ or 48 PgC yr$^{-1}$).

The challenge for the first photosynthetic biosphere would thus have been to evolve the means of recycling the scarce materials that it needed to metabolize, especially the electron donors for photosynthesis. The ease or difficulty of evolving recycling has been examined theoretically by simulating 'virtual worlds' seeded with 'artificial life' forms and leaving the resulting ecosystems to evolve (Williams and Lenton, 2007). In these simulations, the closing of material recycling loops

robustly emerges (Williams and Lenton, 2007), even if they incur an energetic fitness cost (Boyle et al., 2012). The empirical record of how and when recycling emerged in the early Earth system is sparse, but there is some evidence for biogenic methane production by 3.5 Ga (Ueno et al., 2006). This would have recycled hydrogen (and carbon) back to the atmosphere. If the early biosphere was fuelled by anoxygenic photosynthesis based on $H_2$, then recycling of hydrogen via methane production and
photolysis could have boosted global NPP to 1.8 EJ $yr^{-1}$ (48 TgC $yr^{-1}$) or 0.1% of the modern marine biosphere (Canfield et al., 2006). If volcanic activity on the early Earth was elevated by an order of magnitude, a hydrogen-fuelled biosphere might have approached 1% of modern marine NPP (Canfield et al., 2006). Alternatively, if early anoxygenic photosynthesis used the supply of reduced iron upwelling in the ocean then its NPP, controlled by ocean circulation, might have reached 77-225 EJ $yr^{-1}$ (2-6 PgC $yr^{-1}$) or $\sim 10\%$ of modern marine NPP (Canfield et al., 2006; Kharecha et al., 2005) (Fig. 1). A potential constraint
on early biosphere productivity is provided by the carbon isotope record of marine carbonate rocks, which is conventionally interpreted as indicating that the proportion of carbon buried in organic form (rather than inorganic carbonates) was around 20% even as early as 3.5 Ga. Given greater inputs of carbon from the mantle on the early Earth, this would imply a marine organic carbon burial flux in excess of the present 60 TgC $yr^{-1}$, setting a lower limit on NPP at the time (assuming no heterotrophic recycling, i.e. all organic carbon produced was buried). This would likely preclude $H_2$-based photosynthesis as the dominant
source of carbon 3.5 Ga onwards, suggesting instead an iron-fuelled (or even oxygenic) biosphere. However, a more nuanced interpretation of the carbon isotope record allows for the possibility that little organic carbon was buried for large parts of the Archean Eon (Schwartzman, 1999; Bjerrum and Canfield, 2004).

The waste products of early metabolisms would have altered the environment. The long-term burial of organic carbon, even if it was a small flux, would have removed carbon from the atmosphere (and ocean) tending to cool the planet. This cooling
effect could have been profound given that today 15 ZgC are stored as organic carbon in sedimentary rocks, compared to 38 EgC in the ocean-atmosphere system. Somewhat counterbalancing the net removal of carbon to the crust, the conversion of atmospheric $CO_2$ to methane would have increased radiative forcing, tending to warm the planet. As a crustal reservoir of reduced carbon accumulated in sedimentary rocks, some organic carbon would later be exposed on the continents, potentially supporting heterotrophic productivity there. Relatively low estimates of global productivity make it unlikely that the macro-
nutrients nitrogen and phosphorus became limiting, making them under-tapped resources in the ocean environment.

## 2.2  Oxygenic photosynthesis

The next major revolution in energy input to the biosphere was the origin of oxygenic photosynthesis, using water as an electron donor (Lenton and Watson, 2011). To split water requires more energy (i.e. more high energy photons of sunlight) to be captured than in any of the earlier anoxygenic forms of photosynthesis. It was contingent on the prior origin of anoxygenic
photosynthesis in that two existing photosystems – derived from anoxygenic photosynthetic ancestors – were wired together in the same cell (Allen and Martin, 2007). To be naturally selected, oxygenic photosynthesis required an environment – plausibly freshwater (Blank and Sanchez-Baracaldo, 2009) – where easier electron donors were absent or had been drawn down to limiting concentrations. The resulting cyanobacterial cell was the ancestor of all organisms performing oxygenic photosynthesis

on the planet today. It took up to a billion years to evolve (Lenton and Watson, 2011), with the first evidence of oxygen appearing 3.0-2.7 Ga (Farquhar et al., 2011; Planavsky et al., 2014).

Once oxygenic photosynthesis evolved, the productivity of the biosphere was no longer restricted by the supply of substrates for photosynthesis, as water and carbon dioxide were abundant. Instead, the availability of nutrients, notably nitrogen and phosphorus, would have become the major limiting factors on global productivity – as they still are today. Oxygenic photosynthesis would have flourished wherever nutrients were available and anoxygenic photosynthesis drew down its electron donors to limiting concentrations, or where oxygen removed those electron donors by oxidizing them. Anoxygenic photosynthesis might have flourished underneath oxygenic photosynthesis in parts of the surface ocean if and when anoxic waters bearing $Fe^{2+}$ extended up into the sunlit photic zone (Johnston et al., 2009), and this would have set up some competition for the nutrients nitrogen and phosphorus.

Constraints on nutrient concentrations in the early ocean are scarce (Planavsky et al., 2010). Nitrogen would initially have been in the form of ammonium (rather than nitrate), but the advent of an oxygen source plausibly triggered the onset of nitrification and denitrification (Godfrey and Falkowski, 2009; Garvin et al., 2009). Nitrification could have produced small pools of nitrate in restricted surface ocean 'oxygen oases' with nitrogen in the form of ammonium elsewhere. Whether denitrification could then have caused nitrogen scarcity (Godfrey and Falkowski, 2009), depends on whether nitrogen fixation had evolved and could counter-balance it (Zhang et al., 2014). Iron and vanadium-based nitrogen fixation were plausibly already widespread (Zhang et al., 2014), although molybdenum-based nitrogen fixation may have evolved later (Boyd and Peters, 2013). Thus phosphorus was probably the ultimate limiting nutrient, as it is today. Lower terrestrial weathering fluxes of phosphorus (relative to present) have been predicted, due to a shift from terrestrial to seafloor weathering to balance the carbon cycle earlier in Earth history, and this would have tended to reduce ocean phosphorus concentration, because seafloor weathering is not a source of phosphorus (Mills et al., 2014). Initial work estimated only $\sim 10-25\%$ of today's phosphorus concentration in the Late Archean ocean (Bjerrum and Canfield, 2002), however subsequent studies have revised this upwards to $\sim$1-4 times present-day phosphorus concentration (Planavsky et al., 2010). Furthermore, nutrient recycling by the microbial loop within the surface ocean (Azam et al., 1983), was conceivably more efficient than today because eukaryotic mechanisms of exporting organic matter out of the surface ocean were absent. One model suggests that marine NPP may have been $\sim 25\%$ of today's productivity (450 EJ yr$^{-1}$ or 12 PgC yr$^{-1}$) in the Late Archean $\sim 2.7$ Ga (Goldblatt et al., 2006).

With the advent of oxygenic photosynthesis there was thus an order-of-magnitude increase in organic carbon production (Fig. 1). The extra flux of carbon sinking into the anoxic depths of the ocean would initially have fuelled methanogenesis (as sulphate was yet to build up significantly in the ocean (Crowe et al., 2014; Zhelezinskaia et al., 2014)). The resulting upward flux of methane could support widespread methanotrophy near the source of oxygen from oxygenic photosynthesis, consistent with very isotopically-light organic carbon from $\sim 2.7$ Ga (Hayes, 1994; Eigenbrode and Freeman, 2006; Daines and Lenton, 2015). A large flux of methane, equivalent to around 60% of the primary production sinking out of the surface layer of the ocean (Daines and Lenton, 2015), would also escape to the atmosphere, warming the planet. However, if the $CH_4:CO_2$ ratio in the atmosphere approached 0.1, photochemical production of an organic haze that scattered sunlight back to space would have

triggered cooling (Haqq-Misra et al., 2008). This process would be self-limiting, but might help explain the earliest glaciations $\sim 2.9$ Ga (Domagal-Goldman et al., 2008).

Oxygen remained a trace gas, $O_2 < 10^{-5}$ PAL (present atmospheric level), until 2.45 Ga as indicated by the mass independent fractionation of sulphur isotopes (MIF-S), preserved in sediments older than this, which shows that the ozone layer was absent and high energy ultraviolet radiation reached the surface (creating the signal), and sulphate had yet to accumulate in the ocean (allowing the signal to be preserved) (Lenton and Watson, 2011). Elevated concentrations of methane in such a reducing atmosphere would have supported an increased flux of hydrogen loss to space, causing the long-term oxidation of the surface Earth system (Catling et al., 2001). Stability broke down 2.45–2.3 Ga in the 'Great Oxidation' event (Lenton and Watson, 2011). The MIF-S signature disappeared indicating that oxygen rose $> 10^{-5}$ PAL sufficient to form an ozone layer. Massive deposits of oxidized iron appeared in the form of the first sedimentary 'red beds', and oxidized iron also appeared in ancient soils, indicating that oxygen increased to $> 10^{-2}$ PAL. Models suggest that once enough oxygen built up for the ozone layer to start to form, this shielded the atmosphere below from UV and slowed down the removal of oxygen by reaction with methane (Goldblatt et al., 2006; Claire et al., 2006; Daines and Lenton, 2015). This created a strong positive feedback explaining the abruptness of the Great Oxidation (Goldblatt et al., 2006; Claire et al., 2006; Daines and Lenton, 2015).

The Great Oxidation destabilized other environmental variables. As oxygen rose, atmospheric methane concentration declined (Goldblatt et al., 2006; Claire et al., 2006; Daines and Lenton, 2015), which could help explain the series of Huronian glaciations (Haqq-Misra et al., 2008) and the low-latitude Makganyene glaciation 2.32-2.22 Ga (Teitler et al., 2014; Kopp et al., 2005). The reaction of oxygen with sulphide in continental rocks plausibly produced sulphuric acid that dissolved phosphorus out of apatite inclusions in the rocks and fuelled marine productivity (Bekker and Holland, 2012). The oxidizing power unleashed in the Great Oxidation could thus have made another limiting resource, phosphorus, more available, boosting energy input to the biosphere. One model estimates that marine NPP in the Proterozoic Eon after the Great Oxidation was $\sim 1300$ EJ $yr^{-1}$ (34 PgC $yr^{-1}$) or $\sim 70\%$ of today's value (Mills et al., 2014). This would have supported increased organic carbon burial, which is inferred to have occurred during the 'Lomagundi' carbon isotope excursion, 2.23–2.06 Ga (Bjerrum and Canfield, 2004), potentially triggering an 'overshoot' of atmospheric oxygen (Bekker and Holland, 2012; Canfield et al., 2013). However, there were large crustal reduced sinks for oxygen at the time (Bachan and Kump, 2015), and after $\sim150$ Myr, excess buried organic carbon was recycled through the crust back to the surface, consuming oxygen and deoxygenating the ocean (Canfield et al., 2013). After this protracted interval of instability, the Earth entered an even longer period of stability, known as 'the boring billion'.

Marine productivity during this protracted interval of the Proterozoic Eon is very uncertain. We know that the deep ocean remained largely anoxic and 'ferruginous' (with $Fe^{2+}$ in solution), with euxinic waters ($SO_4$-reducing) at intermediate depths along some ocean margins, and surface waters largely oxygenated (Poulton et al., 2010; Planavsky et al., 2011). Several authors have argued for very low productivity partly on the grounds of a sparsity of organic carbon rich shales, but largely based on theoretical arguments for low nutrient conditions. Phosphate supply to the ocean could have been reduced by scavenging onto iron oxides forming in freshwater and estuarine environments (Laakso and Schrag, 2014). Phosphate could also have been efficiently removed from ocean waters by the formation of mixed $Fe^{2+}/Fe^{3+}$ compounds such as 'green rust' (Zegeye et al., 2012).

However, reducing deeper waters and sediments (especially euxinic ones) should have been effective at recycling phosphorus and shuttling it back to the surface ocean, consistent with high estimates of phosphate concentration at 1.7 Ga (Planavsky et al., 2010). Nitrogen limitation has been argued for on the grounds of a lack of molybdenum for nitrogen fixation (Anbar and Knoll, 2002), but the existence of alternative nitrogenases makes this unlikely (Zhang et al., 2014). Instead, heterogeneous ocean redox conditions could have supported a mixed nitrogen cycle with ammonium in the predominantly reducing waters of the deep ocean and small reservoirs of nitrate in oxygenated waters. In such a system there would be large fluxes of denitrification along the extensive interfaces between oxygenated and anoxic waters, counterbalanced by large fluxes of nitrogen fixation in surface waters replenishing the nitrogen reservoirs. Indeed the fact that the Great Oxidation was never reversed sets a lower bound on Proterozoic productivity of $\sim$25% of modern marine NPP in an existing model (Goldblatt et al., 2006).

The Great Oxidation increased energy consumption by the biosphere, even with no change in energy input, because respiring organic matter with oxygen (2870 kJ mol$^{-1}$) yields an order of magnitude more energy than breaking it down anaerobically (e.g. 232 kJ mol$^{-1}$ for alcohol fermentation). This greater energy source facilitated the evolution of new levels of biological organization, in the form of eukaryotes. The ancestral (heterotrophic) eukaryote is thought to have had mitochondria performing aerobic respiration. The timing of eukaryote origins is deeply uncertain, but with putative biomarker evidence 2.7 Ga now rejected (Rasmussen et al., 2008; French et al., 2015), and molecular clocks suggesting a last common ancestor 1.8–1.7 Ga (Parfrey et al., 2011), they may post date the Great Oxidation. Mitochondrial respiration in turn allows eukaryotes to support a much larger genome than prokaryotes, giving them the capacity to create more complex life forms with multiple cell types (Lane and Martin, 2010), the first evidence for which appears $\sim$1.2 Ga (Parfrey et al., 2011; Butterfield, 2000; Knoll et al., 2006).

## 2.3 Eukaryotic photosynthesis and land colonisation

The next revolution in energy input to the biosphere involved encapsulating an existing metabolism – oxygenic photosynthesis – in progressively more complex, eukaryotic organisms and symbioses – algae, lichens and land plants (with mycorrhizal fungi). This energy revolution involved increasing the supply and utilization of limiting nutrient resources needed to perform photosynthesis and increasing the area over which it occurred.

The lineage containing all extant photosynthetic eukaryotes arose 1.7-1.4 Ga (Parfrey et al., 2011), but eukaryotic algae only became ecologically significant relative to cyanobacteria $\sim$ 740 Ma, when biomarkers of algae become more prevalent in ocean sediments and the diversity of eukaryote fossils starts to increase (Knoll et al., 2006). Larger eukaryote cells are better at exploiting excess nutrients in polar surface oceans, but would also have removed carbon and nutrients from the surface ocean more efficiently, thus reducing recycling – with uncertain overall effects on productivity (Lenton et al., 2014). More efficient carbon export to sediments plausibly increased phosphorus removal from the ocean, lowering global productivity and tending to oxygenate the deep oceans (Lenton et al., 2014), and contributing to $CO_2$ drawdown and global cooling (Tziperman et al., 2011). $CO_2$ drawdown by silicate weathering might have been enhanced by the arrival of eukaryotes (fungi and algae) in microbial ecosystems on the land (Lenton and Watson, 2004). Estimates of the productivity of global microbial mats, based on a simple area-scaling of modern desert crust (Brostoff et al., 2005), suggests only 3–11% of today's terrestrial NPP, comparable

to today's cryptogamic cover, which achieves 1–6% of terrestrial NPP (Porada et al., 2013). However, deserts are unproductive environments and modern cryptogamic cover is living in a world dominated by vascular plants. Taking the ecophysiological model of cryptogamic cover (Porada et al., 2013) and considering higher atmospheric $CO_2$ and lack of competition from vascular plants, putative Neoproterozoic-early Paleozoic land biota might have achieved $\sim 25\%$ of today's terrestrial NPP.

Whatever the cause(s), the Earth experienced two low-latitude 'snowball Earth' glaciations – the Sturtian (starting 715 Ma) and Marinoan (ending 635 Ma) – amidst a protracted interval of instability in the global carbon cycle. Iron formations deposited during these events apparently record high concentrations of phosphate in the ocean (Planavsky et al., 2010), which could be explained by a shutdown of biological uptake and removal to sediments. In the aftermath of glaciations, high productivity could have been fuelled – at least temporarily – by elevated phosphorus concentrations. There is evidence of at least partial

oxygenation of the deep ocean, and complex eukaryotic life including animals began to flourish in the oceans. However, by the early Phanerozoic, phosphorus concentrations were broadly comparable to today (Planavsky et al., 2010; Bergman et al., 2004), implying comparable levels of marine NPP.

The key change in energy input to the biosphere and material cycling came later with the rise of plants on land, starting around 470 Ma and culminating in the first global forests by 370 Ma (Kenrick et al., 2012). This roughly doubled global

NPP, increasing it by an order-of-magnitude on land and potentially indirectly in the ocean. Terrestrial NPP is estimated to have exceeded today's value ($\sim 2100$ EJ yr$^{-1}$ or 56 PgC yr$^{-1}$) on average during the Phanerozoic (Bergman et al., 2004) (Fig. 1), with peaks potentially exceeding twice the present value (Beerling, 1999). To colonize the land required new nutrient acquisition mechanisms, achieved through symbioses with mycorrhizal fungi and nitrogen-fixing bacteria. Plants and their associated mycorrhizal fungi accelerated the chemical weathering of the land surface in search of rock-bound nutrients, notably

phosphorus. Ultimately, stunningly effective recycling developed, such that the average terrestrial ecosystem today recycles phosphorus $\sim 50$ times through primary production before it is lost to freshwaters (Volk, 1998).

Increased silicate weathering lowered atmospheric $CO_2$ levels, plausibly triggering the Late Ordovician glaciations (Lenton et al., 2012), although others question the magnitude of early plant effects on the carbon cycle (Quirk et al., 2015; Edwards et al., 2015). A more established view is that the weathering effects of later plants, notably the first deep-rooting trees forming

forests, caused the later Permian-Carboniferous glaciations. Increased phosphorus weathering supplied nutrient to the oceans, increasing marine productivity and plausibly triggering oceanic anoxic events (Algeo and Scheckler, 1998). The increase in organic carbon burial with the rise of plants also increased atmospheric oxygen, as revealed in the charcoal record (Scott and Glaspool, 2006). Although ignition sources (lightning, volcanoes) have always existed on Earth, there was little to burn before land plants arose, and experiments show that $O_2 > 15\%$ of the atmosphere is required for biomass combustion to be

sustained (Lenton and Watson, 2011; Belcher and McElwain, 2008). The first charcoal evidence for natural fires coincides with the appearance of vascular plants on drier land $\sim 420$ Ma (Scott and Glaspool, 2006). Plants in turn provided a new source of organic carbon for burial in sediments, especially new structural carbon polymers (including lignin), which are hard to biodegrade. Fungi evolved to recycle these, but a delay may have caused atmospheric $O_2$ to peak in the Carboniferous (Robinson, 1990; Floudas et al., 2012). The continuous charcoal record indicates $O_2$ persistently >15% of the atmosphere

since 370 Ma (Belcher and McElwain, 2008; Scott and Glaspool, 2006).

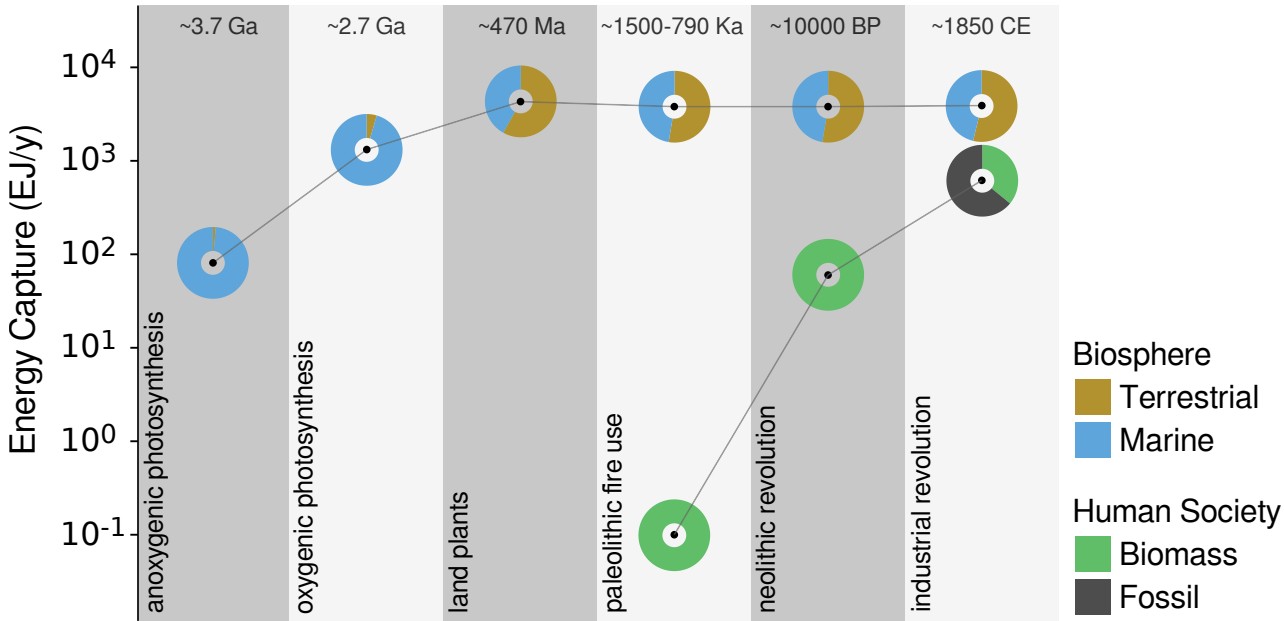

**Figure 1.** Energy capture in the biosphere and human society. Dates indicate beginning of the respective revolution, energy estimates are given for dates were energy regimes had matured. Data and sources in Table S1.

The rise in atmospheric oxygen and increase in food supply brought about by land plants has allowed a flourishing of animal complexity from aerobic pathways – including the emergence of us humans. Today, the total global energy flux through heterotrophic biomass, based on a 10% conversion efficiency of 100 PgC $yr^{-1}$ with energy density 40 kJ $gC^{-1}$, is $\sim 400$ EJ $yr^{-1}$, roughly half on land and half in the ocean. Natural fires additionally consume $\sim 55$ EJ $yr^{-1}$ (1.4 PgC $yr^{-1}$) (Eliseev et al.,
5   2014), and human-induced fires $\sim 45$ EJ $yr^{-1}$ (1.1 PgC $yr^{-1}$) (Haberl et al., 2007), giving a total biomass burning flux today of $\sim 100$ EJ $yr^{-1}$ ($\sim 2.5$ PgC $yr^{-1}$) (Randerson et al., 2012), or $\sim 2.5\%$ of the energy and carbon captured in photosynthesis.

## 3   Revolutions in human history

Like all animals humans are heterotrophs. Our biological metabolism relies on the products of photosynthesis. At the same time humans are exceptional among animals in creating and maintaining a social metabolism via breeding and cultivating
10   plants and animals, in constructing buildings and large infrastructure systems and in producing numerous artifacts (Ayres and Simonis, 1994; Fischer-Kowalski, 1998; Weisz et al., 2001). The social metabolism inevitably extends total human energy capture and material use beyond the biological requirements. In modern industrial societies the amount of energy and materials used to produce and reproduce domesticated livestock and all artifacts typically is two orders of magnitude larger than the basic biological metabolism of the human population itself. For the following comparison between human energy use and the
15   primary productivity of the entire biosphere, it is therefore important to keep in mind the different trophic levels involved,

autotrophs versus heterotrophs, and the unique capability of human societies to extend their biological means of energy and materials utilization through agriculture and technology.

A critical question in this regard is how to define the system boundary of human society vis a vis its environment in terms of inputs and outputs of energy and materials. For materials we apply the method implemented by the European Statistical Office (Fischer-Kowalski et al., 2011; Krausmann et al., 2015). According to this method all raw materials, except water and air, that serve the production and reproduction of humans, livestock, buildings, built infrastructure, durable and non-durable goods and services are accounted for as socioeconomic input. The main raw material inputs to modern societies are therefore plant harvest for food, feed, other energy uses and as material input to industrial production, sand, gravel and crushed stone mainly for construction purposes, metals and non-metallic minerals for various industrial production purposes, and fossil energy carriers for both energetic and material applications. The national indicator derived from this method is domestic material consumption (DMC) defined as raw materials extraction plus imported goods minus exported goods measured in tons per year(Weisz et al., 2006).

Regarding energy we deviate from the most common approach to account for total primary energy supply (TPES) used in national and international energy statistics, see e.g. the annually published energy balances by the International Energy Agency. TPES excludes plant biomass used for food and feed which makes this indicator unsuitable for a comprehensive reconstruction of the evolution of human energy use in a deep history perspective. Instead, the method used here applies the same system boundary to the material and the energetic dimension, taking into account the primary energy used in technical conversion processes as well as the energy content of plants for human nutrition and for feeding domesticated animals (Haberl, 2001).

Energy capture by human societies involves trophic levels and specific mechanisms which are different from those occurring during primary production at the planetary scale. A comparison between the two is still warranted, as human society inevitably operates within the thermodynamically closed Earth system. The emergence and continued existence of human civilization is conditional upon the stability of certain basic dynamics of the Earth system which are vulnerable to metabolic changes in kind and scale, such as changes in the overall energy balance, or changes in the chemical composition of the atmosphere, oceans or soils, rather than the specific mechanisms that caused them.

## 3.1 Paleolithic fire use

During most of their existence humans lived as foraging societies in an uncontrolled solar-energy system (Sieferle, 1997), simply tapping into the existing energy and material cycles of the biosphere, without deliberately controlling them by systematic land management, and without introducing new biogeochemical pathways. The first human revolution in energy input was the intentional use of fire, which set humans apart from all other species. With it humans extended their energy utilization beyond their biological metabolism towards areas outside the human body. This marked the beginning of a social metabolism – a collectively organized extension of energy and material use by human societies (Fischer-Kowalski, 1998; Fischer-Kowalski and Weisz, 1999).

There is robust evidence that *Homo erectus* could control fire from 790 ka in Africa (Pausas and Keeley, 2009) and from 400 ka in Europe (Roebroeks and Villa, 2011). The ability to cook, which implies the control of fire, may date as far back as 1.5 Ma

(Wrangham et al., 1999). Cooking provided higher food energy, higher food diversity through detoxification, and a selective force to develop social abilities and large brains, thus playing a key role in human evolution. The use of fire may also have facilitated humans occupying colder climates (Gowlett, 2006), and developing increased abilities to cooperate (Brown et al., 2009), a decisive element of their evolutionary success.

Use of fire for cooking increased energy input to approximately 7–15 GJ cap$^{-1}$ yr$^{-1}$, i.e. a factor of 2–4 above the average physiological energy demand of 3.5 GJ cap$^{-1}$ yr$^{-1}$ (Simmons, 2008; Fischer-Kowalski and Haberl, 1997; Boyden, 1992). Assuming a population of 2–4 million at the beginning of the Neolithic (Cohen, 1995), overall energy capture by humans amounted to roughly 14–60 PJ yr$^{-1}$, a factor of $\sim 1000$ below the global human energy input in 1850 and $\sim 10,000$ below today's (Fig. 1). In foraging societies, biomass accounts for more than 99% of material input. Materials are used predominantly
for energetic purposes, as fire wood or food. Thus the energetic and the material social metabolism were practically identical.

Based on their direct energy and material inputs, foraging societies had a negligible impact on the global environment. However, the intentional use of fire for hunting, clearing land and other purposes could have caused significant environmental impacts – accepting that the empirical evidence regarding frequency, scale and age for applying those intentional burning techniques is highly contested. Potential impacts include extinction of large Pleistocene land animals and ecosystem tipping events,
including shift of vegetation to desert shrub triggering a weak monsoon in Australia (Miller et al., 2005), rapid landscape transformations in the mesic environments of New Zealand (McWethy et al., 2010), the wet tropical forests of the pre-Columbian Amazon (Nevle et al., 2011), and across the savannas and woodlands of Africa (Archibald et al., 2012).

Foraging societies need large areas. Although the energy density of natural vegetation ranges over 0.1-1 W m$^{-2}$ (NPP of 3.16-31.6 MJ m$^{-2}$ yr$^{-1}$) (Smil, 2008), the bulk biomass of the most abundant plants, grasses and trees, is not edible for humans.
The very small share of human-edible natural biomass restricts the population density of foraging societies to no larger than $\sim$ 0.02$-$0.2 cap km$^{-2}$ (Simmons, 2008). Such low population densities and the necessity to stay mobile prevent the accumulation of artifacts and the development of complex institutions – e.g. institutions to deal with conflict are prohibitively costly as long as moving away is an attainable alternative. Therefore foraging societies are typically portrayed as small egalitarian groups of low internal complexity, based largely on a few extant foraging societies who have been pushed aside to marginalized environments.
In more favorable environments higher resource intensities could have supported higher population densities and significantly more complex social structures, including settlements, handcraft, trade and social stratification(Headland et al., 1989; Gowdy, 1997).

## 3.2  The Neolithic revolution

By the beginning of the Holocene, 11,700 BP, humans had successfully inhabited all continents. Then, within a few thousand
years a fundamentally new socio-metabolic energy regime emerged on all continents except Australia, involving the domestication of wild plant and animal species and the control of their reproduction via husbandry. Agriculturalists greatly enhanced the area productivity of edible species at the expense of non-edible species and of food competitors. In contrast to pre-agricultural societies they lived in a controlled solar energy system (Sieferle, 1997).

Agriculture had multiple independent origins; in the Near East (∼10000 BP), Peru (∼10000 BP), South China (8500 BP), North China (7800 BP), Mexico (4800 BP), East North America (4500 BP), and possibly sub-Saharan Africa (4000 BP) (Smith, 1995; Dillehay et al., 2007; Diamond and Bellwood, 2003; Barker, 2006). Archaeological evidence from several sites at the shoreline of the Persian Gulf has revealed a rapid colonization of this area by advanced agricultural and urban societies at

around 7500 BP. As sea-level rise from the last glacial low stand was only completed in the Persian Gulf 7000-8000 BP, there could be even older agricultural sites in areas that are now beneath the Indian Ocean (Rose, 2010). Explaining the relatively rapid transition to agriculture is one of the most controversial topics in universal history. The puzzle is that early agriculture, especially farming, was not obviously superior to foraging. Ethnological studies have shown that early farmers spent more hours to exploit their food base, relied on a less diverse and less stable diet, were more prone to diseases, and even had

lower productivity in terms of calorific return on labor investment (Boserup, 1965; Bowles, 2011). The Neolithic revolution therefore tends to be explained as a necessity driven transition, fostered by population pressure (Boserup, 1965), deterioration of resources (extinction of Pleistocene megafauna), or climate change. Whatever the reasons for switching from foraging to farming, it creates a lock-in once population densities exceed the natural carrying capacity of the surrounding ecosystem. Then reverting to foraging cannot occur without substantially reducing population numbers.

After ∼ 7000 BP complex agrarian civilization emerged (Sieferle, 1997). Extant biomass was still the energy source for almost all energy uses: food, fodder, heat, mechanical power and chemical transformation (metallurgy). Wind and water power used by agrarian civilizations (sailing ships and mills) were locally important but contributed only marginally to the energy input. Despite huge variations in agrarian land use systems, a defining condition is that energy supply is tightly coupled to productive land and (human and animal) labor working on the land. Without any external energy subsidies in the form of

mechanical power and synthetic fertilizers, a larger usable energy output from extant biomass typically requires more land or more labor input on existing land, thus putting relatively strict limits to the possibility of increasing energy supply per capita (Krausmann et al., 2008b). Higher yields could be achieved by various improvements in agricultural technology but those improvements were typically population driven and lead to absolute growth in energy capture per area of land while per capita energy availability stagnated or even declined (Boserup, 1965). Estimates of global average energy input to agrarian

societies are 45-75 GJ cap$^{-1}$ yr$^{-1}$ roughly a factor of 5 greater than in foraging societies (Fischer-Kowalski et al., 2014). With estimated population rising to ∼ 450 million in AD 1500 (when the agrarian mode of subsistence dominated the global population), overall energy capture by humans may have reached ∼ 20 EJ yr$^{-1}$ (Fischer-Kowalski et al., 2014), a factor of 300 above the foraging regime, but 30 below today. When the industrial revolution took off around 1850 human population was ∼ 1.3 billion and energy capture had reached ∼ 60 EJ yr$^{-1}$(Fig. 1).

The increased population and energy flows due to farming increased the material inputs to, and waste products from, societies. The resulting environmental effects began early in the Holocene, but their scale is much debated (Ellis et al., 2013; Ruddiman, 2013). Irrigation began around 8000 BP in Egypt and Mesopotamia, leading to some localized salination and siltation of the land, reducing crop yields and encouraging a shift in agricultural crop from wheat to more salt-tolerant barley (Jacobsen and Adams, 1958). The use of manure as fertilizer may have begun as early as 9000 BP in SW Asia and 7000 BP in

Europe (Ellis et al., 2013; Bogaard et al., 2007). The clearance of forests to create agricultural land and supply biomass energy

and wood from 8000 BP onwards, reduced the carbon storage capacity of the land, transferring $CO_2$ to the atmosphere (Kaplan et al., 2011). Cumulative carbon emissions may have approached 300 PgC by 500 BP (Kaplan et al., 2011) contributing $\sim 20$ppm to atmospheric $CO_2$ levels. The biogeophysical effects of forest clearance also affected the climate, regionally and remotely (Devaraju et al., 2015). Anthropogenic sources of methane started around 5000 BP with the irrigation of rice paddies and have contributed to changes in atmospheric $CH_4$ concentration over the past $\sim 3000$ years (Mitchell et al., 2013).

The energetic surplus generated by agrarian societies first allowed cities to become a widespread phenomenon $\sim 5000$ years after the beginning of agriculture. This led to more complex social organization with increasing division of labor, technological innovations, social stratification and written language (Sieferle, 1997). This in turn requires re-integration via exchange, trade and redistribution creating mutual dependencies which increased the potential for conflict, prompting the inception of social institutions to deal with such conflicts (e.g. priests, judges). Additionally, stockpiling and concentration of resources in cities attracted predators stimulating institutions of defense (military). Social complexity has costs as well as benefits and a number of early complex societies collapsed when those costs became prohibitively high (Tainter, 1988).

Agrarian societies are faced with relatively severe constraints regarding the energy surplus they can achieve. On average around 90% of the population is required to work in agriculture. This limits the urban population engaged in non-food producing activities to no more than 10% (GEA, 2012), although locally urbanization levels could be much higher. The outstanding role of bio-productive land as the main factor of production also explains the important political role of territory in agrarian societies. The intrinsic connection between social stratification and territory in agrarian societies can be illustrated by the role of land in medieval European feudalism, where the power of the nobility was strongly connected to the control over productive land. Economic growth was only possible through land expansion and increase in area productivity. Both have inherent practical limits and both require a growing population. Combined with hard constraints on transportation in pre-industrial societies, this leads to relatively fast local negative feedbacks in the energetic and material social metabolism and renders sustained material growth impossible on a per capita basis – making the distribution of material wealth a zero sum game.

### 3.3 The Industrial revolution

Fossil fuels, especially coal and peat had been used for hundreds of years in China, Burma, The Netherlands and England (Ayres, 1956). However, their contribution to the social metabolism always remained small. The key energy transformation of the industrial revolution came with the ability to massively scale-up fossil energy use (Sieferle, 1997, 2001; Wrigley, 2010). Unlike the Neolithic revolution, the industrial revolution was a historical singularity. Its inception in 18th century England was followed by a worldwide expansion of the new energy regime, which is still ongoing. The fossil energy regime eventually surmounted the inherent thermodynamic constraints of agrarian societies that had existed for millennia by decoupling socially usable energy from bio-productive land and human labor (Krausmann et al., 2008b). Within 150 years, from 1850 to 2000, global human energy use increased tenfold from 56 EJ $yr^{-1}$ to 600 EJ $yr^{-1}$ (estimates based on (Krausmann et al., 2008b; Fischer-Kowalski et al., 2014)), the world population (van Zanden, 2015) went from 1.3 billion to 6 billion, and global GDP (TMP, 2015) increased from 800 to 6,600 intGK\$. Thus by 2000 the annual global energy flux through human societies was one

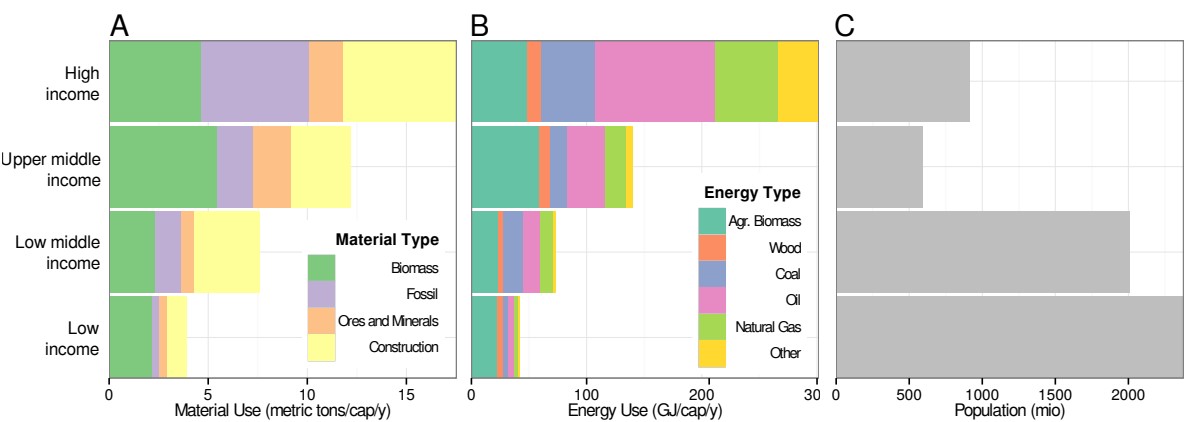

**Figure 2.** Year 2000 (A) material (Steinberger et al., 2010) and (B) energy use per capita (Krausmann et al., 2008a) and (C) total population (data.wordbank.org), by income groups.

third of the global terrestrial NPP (Haberl et al., 2007) and one third above the total global energy flux through all non-human heterotrophic biomass (Fig. 1).

Unlike the Neolithic revolution, the puzzle of the industrial revolution is not that it began, but that it continued (Wrigley, 2010). Similar innovation driven growth periods in agrarian civilizations (e.g. the Dutch golden age) could not be sustained,
because they were sooner or later counterbalanced by diminishing returns on energy investment in the agricultural sector. Even for the classical British economists Adam Smith, David Ricardo, Thomas Malthus, and John Stuart Mill, who witnessed England's industrial take-off, there was no doubt that diminishing marginal yields in the agricultural sector would eventually bring industrialization to a halt (Sieferle, 2010). A key challenge was to feed a growing industrial labor force with a controlled solar-energy based system of agriculture (given that the agricultural sector did not industrialize until the 1930s in the USA and
the 1950s in Europe) (Krausmann et al., 2008b). England was in a specially favored position, because since the late 16th and early 17th century area yields, total agricultural production and labor productivity had been growing continuously (Broadberry et al., 2015). This allowed 18th century England to support a growing industrial labor force in the initial phase of the industrial revolution. When agricultural productivity gains eventually came to a halt around 1830 – while the population was still growing rapidly – England's hegemonic political position was instrumental to massively increase food imports (Krausmann et al., 2008b;
Broadberry et al., 2015).

The availability of technologies to overcome bottlenecks in energy utilization also played a decisive role in the industrial revolution happening in England. Notably, the coincidence of a domestic endowment of coal with the emergence of a new technology complex consisting of the steam engine and coke based iron smelting. With this technological complex energy constraints could be exceeded (Grubler, 2004), which had previously limited coal extraction, steel production, and long distance
transportation.

The step increase in energy capture with industrialization is associated with fundamental changes in global material cycles. Material inputs to societies were transformed from biomass dominance to minerals dominance. Global average per capita

material use increased from 3.4 to 10 t $cap^{-1}$ $yr^{-1}$ from 1870 to 2000, and with roughly constant biomass use of 3 t $cap^{-1}$ $yr^{-1}$, the average use of mineral and fossil materials increased from 0.4 to 7 t $cap^{-1}$ $yr^{-1}$ (Krausmann et al., 2013b, 2009). In industrial economies $\sim 80\%$ per weight of the total annual outflow of materials is $CO_2$, making the atmosphere the largest waste reservoir of the industrial metabolism (Matthews et al., 2000). Between 1850 and 2000 global $CO_2$ emissions from combustion of fossil fuels and materials processing increased 125-fold from 54 to 6750 TgC $yr^{-1}$ and reached 9140 TgC $yr^{-1}$ in 2010 (Marland et al., 2007).

Industrial societies require large physical stocks: buildings, transport infrastructure, energy, water and waste infrastructure, production facilities and durable consumer goods. For example, the material stock of industrializing Japan has increased by a factor of 40 between 1930 and 2005 reaching 38.7 billion tonnes or 310 tonnes per capita (Fishman et al., 2014) and the non-metallic minerals incorporated in residential buildings, roads and railways in the EU25 was 75 billion tonnes or 203 tonnes per capita in 2009 (Wiedenhofer et al., 2015). In the USA the amount of iron incorporated in durable products and infrastructure increased from 100 million tonnes to $\sim 3200$ million tonnes between 1900 and 2000 (Müller et al., 2006). Industrial societies also use a much larger diversity of minerals. Almost all metals are now commercially used in increasingly complex combinations (Graedel et al., 2015). Overall recycling rates (measured as the global average of the content of secondary metal in the total input to metal production) of metals are uncertain. Recycling rates are above 50% for only three metals (Nb, Ru, Pb), between 20-50% for another 16, and below 20%, often less than 1%, for all the other $\sim 40$ metals in wide industrial use (UNEP, 2011). A recent study estimated that only 6% of globally extracted materials are currently recycled within the socioeconomic system (Haas et al., 2015).

The global biogeochemical cycles of nutrients have also been transformed by industrialization. Between 1860 and 2005 anthropogenic creation of reactive nitrogen grew more than tenfold, from $\sim 15$ to 187 TgN $yr^{-1}$ (Galloway et al., 2008). Furthermore, the creation of nitrogen oxides as a waste product of fossil fuel combustion increased from $\sim 0$ to 25 TgN $yr^{-1}$ (Galloway et al., 2008). The excess reactive nitrogen was transferred to other environmental pools, partly denitrifying to atmospheric $N_2$, but also contributing to eutrophication and acidification of terrestrial and coastal marine ecosystems, to global warming and to tropospheric ozone pollution. Analogous human induced acceleration affected the P-cycle.

The industrial revolution also gave rise to entirely new metabolites. The CAS Registry (ACS, 2015) currently includes 92 million unique chemical substances in commercial use of which only 320,000 are regulated in key markets. It is unknown how many of these substances represent entirely new chemicals and whether they are harmful to humans or the environment. With 15,000 new entries daily comprehensive in-vivo toxicity testing is practically impossible (Rovida and Hartung, 2009).

The industrial revolution expanded to the European continent and to the USA in the early 19th century, to Japan in the late 19th century and to large nations like China, India, and Brazil in the last decades of the 20th century. With the transition to an industrial mode of production the socio-economic power of the nobility (based on control over productive land) diminished and shifted towards the owners of the means of industrial production (which Karl Marx called capitalists). Large differences in consumption among countries persist until today (GEA, 2012) (Fig. 2; data from (Krausmann et al., 2008a; Steinberger et al., 2010) and the World Bank Income Classification (WB, 2015)). If we consider high income countries with an average energy use of 302 GJ $cap^{-1}$ $yr^{-1}$ as fully industrial, and upper middle and lower middle countries, with an average energy use of

140 and 74 GJ $cap^{-1}$ $yr^{-1}$ respectively as transitioning to an industrial energy regime, then $\sim 15\%$ of the world population lived in a mature industrial energy regime in 2000, $\sim 44\%$ were in transition, and the remaining $\sim 40\%$ still lived under largely agrarian conditions with average energy use amounting to 42 GJ $cap^{-1}$ $yr^{-1}$. The correlation between energy use and human development appears to be highly non-linear. At high levels of human development large increases in energy input have little or no effect on further increases in standards of living. However, at low levels of human development relatively small increases in energy input have large positive effects (Steinberger and Roberts, 2010), for example supplying $\sim 3.5$ kilowatt per person can greatly increase life expectancy (Schwartzman and Schwartzman, 2013).

## 4 Forward look: A solar powered recycling revolution

Each revolution in Earth and human history involved a new mechanism to capture free energy and the accessing of previously underutilized resources. The resulting step increase in free energy input privileged the systems, biological or social, using the new energy capture mechanism, making them globally significant or even dominant. However, material constraints ultimately became limiting to the expansion of energy innovators either because the the scale of waste products they generated disrupted their environment, or because the material resources they depended upon became scarce. The lesson for human society is that to have a long-term sustainable future within the Earth system will require both a sustainable source of energy and the closure of material cycles (Lenton and Watson, 2011; Weisz et al., 2015; Weisz and Schandl, 2008).

A sustainable energy system is challenging but feasible from a purely technological point of view. The technical potential for renewable energy technologies, most of which ultimately rely on solar energy, exceeds current and future global primary energy demand by several orders of magnitude (GEA, 2012). However, the rate of de-carbonization of the global energy system is constrained by a number of economic (e.g. economic viability of renewable energy technologies, large up-front investments, devaluation of investments in existing energy infrastructure), socio-cultural (e.g. public acceptance of large scale infrastructure projects, food security and various other competing land uses), and technological (e.g. issues of transmission, integration and storage) factors (Fischedick et al., 2011). Current assessments of global development scenarios with ambitious climate mitigation targets put the supply of RE between 250-500 EJ $yr^{-1}$ in 2050 (GEA, 2012; Fischedick et al., 2011; Clarke et al., 2014). Depending on assumptions this corresponds to 25%-75% of the projected (2050) global primary energy demand. The importance of other, more contested energy technologies for achieving a sustainability transition of the global energy system depends on the development of future energy demand. Assuming ambitious energy efficiency improvements the transformation goals can be achieved without nuclear fission, carbon-capture and storage, or high-tech carbon sink management. With less progress on the demand side, one or more of these technologies would be required in the energy mix (Riahi et al., 2012). Nuclear fusion might be an option in the long-term, but is no attainable option in the coming decades when climate mitigation measures must be implemented (World Bank, 2012). Significant additional investments and several decades of technology development would be needed to bring nuclear fusion into large scale practical implementation (von Hippel et al., 2012).

Whilst energy generation for (post-) industrial purposes can be largely de-carbonized, food energy production cannot. The carbon cycle linked to food production can conceivably be re-closed, through a combination of reductions in land-use change

$CO_2$ emissions, and land-based carbon dioxide removal (CDR). However, the much larger (in a fractional sense) perturbations to nutrient (N and P) cycling present a greater challenge, for two contrasting cycles. Nitrogen is abundant in the atmosphere and returned there relatively rapidly by natural biological recycling processes – hence with a sustainable source of energy, nitrogen could be fixed indefinitely. Phosphorus, in contrast, is a rock-bound, finite and non-substitutable resource likely facing either economic (Scholz et al., 2013) or physical scarcity within this century (Van Vuuren et al., 2010). For both nutrients there is a need to minimize the harmful by-products of excess deposition. Yet fertilizer N and (especially) P demand is set to increase significantly with an ongoing shift to more meat-rich diets (Bouwman et al., 2013).

In addition to reversing this trend there is huge potential to counteract this increase through more efficient phosphorus and nitrogen application to crops (through e.g. better targeted fertilizer application), and reducing losses from domestic animal (and human) excrement, crop-residues and the post-harvest life cycle (Clift and Shaw, 2012; Cordell et al., 2011).

The longevity of manufactured capital leads to considerable path dependency and even lock in, and complicates its analysis and accounting. Recent studies have investigated the material stocks of specific metals (Müller et al., 2011) and there are some signs of saturation for specific material stocks in industrialized countries. However, it is unclear to what extent a saturation of the stocks of any single metal are due to material substitution (Fishman et al., 2014). Even if stocks for bulk materials (mainly for construction) were to saturate in industrialized countries due to the projected stabilization of population and slow economic growth, stock levels will need to increase dramatically in emerging and developing countries. Careful design and implementation of these future stocks holds huge potential to slow further growth of the industrial metabolism and minimize lock in.

The explosive proliferation of new metabolites could be tackled by a shift toward *green chemistry* (Linthorst, 2010) that encourages the design of products and processes that minimize the use and generation of hazardous substances. However, given the immense amount of newly introduced chemicals and the importance of material and chemical design for many high-tech produces, additional strategies will be necessary. These may range from new and faster toxicity screening tools, to environmental design guidelines to regulations regarding recyclability and biodegradability material components and final products, applying cradle to cradle principles.

The solar-powered material-recycling "revolution" that we have sketched out demonstrates that the material dimension of the industrial metabolism is much more complex, much more inert and inflexible, and at the same time much less understood than its energetic dimension. Furthermore, such a revolution must anticipate a level of social organization that can implement the changes in energy source and material cycling without preventing present and future generations from attaining similar achievements in standard of living and individual liberation associated with industrial societies. With regards to the lasting attention to Georgescu-Roegen's flawed fourth law of thermodynamics (Georgescu-Roegen, 1971) it is important to note that such a "revolution" does not contradict any established thermodynamic laws (Fleissner and Hofkirchner, 1994; Ayres, 1999; Schwartzman, 2008) as is amply demonstrated by the biological evolution of the Earth's biota.

Pertinent large scale systemic characteristics and relevant regional ramifications of the material social metabolism are still poorly understood, e.g. the structure and dynamic of complex global material supply chains, path-dependency and potential lock-in created by the different components of the manufactured capital, quantitative assessments of the technical and economic

potential to close materials cycles, or effective means to balance the huge number of newly introduced chemicals with feasible tools to assess their toxicity for humans and other species. Furthermore multiple barriers as well as co-benefits between a solar-powered material cycling revolution and other sustainability goals such as climate mitigation, adaptation, reducing extreme poverty, reducing social inequalities, and increasing health are severely under-researched.

5      Future societies might look back at the period of a globally expanding industrial metabolism, with its characteristic exponential material growth, as a necessary phase to transition from the inherently scarce agrarian controlled solar energy system to a second generation controlled solar energy that can provide "affluence without abundance" (Sahlins, 1972) at a much higher level than foraging societies could ever achieve. An outstanding task therefore is to formulate a steady-state "Earth system economics" that supports long-term human and planetary well-being. Two of the most difficult problems to be solved along the

10    way will be to find out how desirable attributes of society, such as knowledge, can still grow while resource input is constrained and how to organize a just distribution of access to physical and non-physical resources in an economy that functions physically as a zero sum game.

*Acknowledgements.* The initial outline and ideas developed in this manuscript were first conceived at the inaugural LOOPS workshop (Chorin, Berlin, February 17-18 2014). TML was supported by a Royal Society Wolfson Research Merit Award.

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
