# Peer review of "Revolutions in energy input and material cycling in Earth history and human history"

_Earth System Dynamics, 2015_

## Referee Comment (RC1) · D. Schwartzman (Referee) · 27 Jan 2016

I really enjoyed reviewing this paper both for its subject and for its comprehensive approach and in-depth scholarship, plus I got a chance to present my own views to those reading this online journal. This paper should be a valuable source for researchers, while the culminating section on a solar powered recycling revolution should be an inspiration to both scientists and activists around the world, a call to make this other world that is possible come into being, before our children and grandchildren are plunged into the abyss of climate catastrophe. Wind and solar power are potentially the energy source of a 21st century industrial revolution, a reprise of the role of coal in the 18th century apparently a result of capital's drive to control labor power versus using its alternative water mills (Malm, 2016).
[Figure]

The paper's underlying theme is the succession of revolutions in Earth history, following Lenton and Watson's (2011) earlier invocation of this concept (see e.g., my review, Schwartzman, 2014a). The authors brilliantly illuminate each revolution, focusing on the stepwise increase in capture of energy by the biosphere (see their Figure 1). But not examined in this paper is whether these revolutions were essentially inevitable products of biospheric evolution, roughly deterministic or rather unique to Earth history. This question is of intense interest to the astrobiology community and its critics. For example, is the emergence of oxygenic photosynthesis very likely, given the likely abundance of water on Earth-like planets around Sun-like stars? I side with the determinists, play the tape again, pretty close to the same outcome (Schwartzman, 1999, 2002; 2015c), a hypothesis likely to be tested in the next few decades by the exoplanetary research program. Further, in view of the order of magnitude billion year lag times in these revolutions, culminating in the very condensed outcomes in human history, were there constraints responsible for holding them back for so long or was this timing simply a matter of chance as Lenton and Watson (2011) argue?

I have pointed out the constraint commonly ignored, the climatic temperature history of the geologic past. While still under debate, I find the oxygen isotope record in sedimentary chert and the compelling case for a near constant isotopic oxygen composition of seawater over geologic time supporting thermophilic surface temperatures prevailing in the Archean. A cold Archean is hard to explain given the likely higher outgassing rates of carbon dioxide, significantly smaller land areas and weaker biotic enhancement of weathering than present in the context of the long-term carbon cycle, taking into account the fainter Archean sun in climate modeling. This evidence points to an important conclusion regarding biological evolution, namely to the critical role of a temperature constraint holding back the emergence of major organismal groups, starting with phototrophs, culminating with metazoans in the latest Precambrian. As a result of the co-evolution of life and its abiotic environment, the evolution of Earth's biosphere is close to being deterministic, i.e., its origin and history and the general pattern of biotic evolution are very probable, given the same initial conditions, potentially a model for
Earth-like planets around Sun-like stars (Schwartzman, 1999, 2002; 2015a).

Briefly mentioned is the role of biotic enhancement of weathering (BEW) in triggering snow ball Earth episodes in the late Proterozoic (lines 274-275 citing Lenton and Watson, 2004). But evidence for the progressive increase in the BEW over geologic time has profound implications for the coevolution of the biosphere and its biota, since BEW represents a powerful catalytic factor in the long-term carbon cycle (Schwartzman and Volk, 1989; Schwartzman, 1999, 2002; 2015b). The Archean (and Hadean) was likely near abiotic with respect to BEW, so this factor must be taken into account in modeling the long-term carbon cycle in the context of abiotic drivers, namely rising solar luminosity, continental area and decreasing volcanic outgassing over geologic time to the present.

Now for revolutions in human history.

Line 567. "The correlation between energy use and human development appears to be highly non-linear. At low levels of human development relatively small increases in energy input have large positive effects, while at high levels of human development large increase in energy input have little or no effect on further increases in standards of living (Steinberger and Roberts, 2010)."

Nevertheless, supplying the rough minimum of 3.5 kilowatt per person to the energy-deprived global South could dramatically increase life expectancies, arguably a robust measure of quality of life (Schwartzman and Schwartzman, 2011; 2013).

Line 570. "Forward look: A solar powered recycling revolution" is a welcome end section to this paper. I have long been advocating the same approach, recognizing that besides avoiding the well-know negative impacts of fossil fuels and and more contentiously nuclear power, high efficiency collection of solar radiation with wind and solar technologies has the capacity to do the work required for recycling. The energy base of the global physical economy is critical: global wind/solar power will pay its "entropic debt" to space as non-incremental waste heat, unlike its unsustainable alternatives
(Schwartzman, 1996, 2008, 2009).

Line 580-590. "A sustainable energy system is challenging but feasible from a purely technological point of view. The technical potential for renewable energy (RE) technologies exceeds current and future global primary energy demand by several orders of magnitude (GEA, 2012). However, the rate of de- carbonization of the global energy system is constrained by a number of economic (e.g. economic viability of RE technologies, large up-front investments, devaluation of investments in existing energy infrastructure), socio-cultural (e.g. public acceptance of large scale infrastructure projects, food security and various other competing land uses), and technological (e.g. issues of transmission, integration and storage) factors (Fischedick et al., 2011). Current assessments of global development scenarios with ambitious climate mitigation targets put the supply of RE between 250-500 EJ yr−1 in 2050 (GEA, 2012; Fischedick et al., 2011; Clarke et al., 2014). Depending on assumptions this corresponds to 25%-75% of global primary energy demand.
"

I find the authors being too conservative in their assessment given the challenges humanity is facing in this century. Going beyond business-as-usual modeling being cited here is imperative. Radical changes in global political economy are likely necessary to implement a timely and robust energy transition to wind/solar power coupled with the elimination of fossil fuels. In thirty years or less this transition could deliver two times the global primary energy consumption level of 551 EJ yr−1 in 2014 (BP, 2015), equivalent to 35 Tera Watts, even with present efficiencies in collection which will increase in this time frame (Schwartzman and Schwartzman, 2011; Schwartzman, 2014). Assuming modest population growth by 2050 corresponding to roughly 9 billion, this level will be needed to terminate energy poverty in the global South, insuring state-of-the-science life expectancy for all of humanity, as well as to generate the incremental energy required for carbon-sequestration from the atmosphere and climate adaptation.

We are now a Type I civilization in Kardashev's scale of cosmic civilizations (Kardeshev, 1964), at a bifurcation, an imminent choice between the collapse of civilization or the
emergence of a truly planetary civilization mobilizing our star's fusion energy for human and nature's needs instead of our present reality of perpetual war on both entities. I submit that an approximate doubling of global energy consumption using this energy source is a necessary condition for the better choice.

Lines 641-644. "An outstanding task therefore is to formulate a steady-state "Earth system economics" that supports long-term human and planetary well-being. One of the most difficult problems to be solved along the way will be to find out how a steady-state society can find new ways to organize a just distribution of wealth in an economy that functions physically as a zero sum game."

Commonly "steady-state" is taken as referring to a no-growth economy. However, the qualitative versus quantitative aspects of economic growth should be distinguished, with the concept of economic growth being deconstructed, particularly with respect to ecological and health impacts. Growth of what are we speaking, weapons of mass destruction, unnecessary commodities, SUVs versus bicycles, culture, information, pollution ? Instead, advocates of global degrowth with their goal of reaching a zero growth steady-state economy commonly lump all growth into a homogenous outcome of the physical and political economy (Schwartzman, 2009, 2012). A zero growth economy is a unwelcome prescription for the immediate challenges posed by the threat of catastrophic climate change as well the undeniable lack of material consumption enjoyed by the majority of humanity living in the global South, the lack of adequate nutrition, housing, education and provision for health services, but most critically, their state of energy poverty. A sustainable growth phase, beginning in capitalism itself, must necessarily have a different quality than capitalist economic growth as measured by the GNP, namely not only requiring global growth in the wind and solar power infrastructure, but also in the agroecological sector. Sustainable economic growth would include global solarization of energy supplies, demilitarization and ecosystem repair (Schwartzman 2009).

More specific comments

Line 89. Update: evidence for biogenic carbon in 4.1 Ga zircons (Bell et al., 2015). Line 131-139. The authors are correct to point out the ambiguity of the sedimentary isotopic record of carbon; also see my discussion in Schwartzman (1999, 2002), p. 26-31. A higher volcanic outgassing rate of carbon in the Archean was balanced by a more intense silicate weathering sink on smaller continents and oceanic islands, plausibly releasing a higher flux of nutrients to the ocean (Schwartzman, 1999, 2002; 2015b). Hence marine biotic productivity may have been enhanced, and biospheric energy capture in the Precambrian may be underestimated. Line 276. Modern desert microbial crust productivity is not a good model for Proterozoic or early Phanerozoic terrestrial biota. Line 416. Even earlier agriculture in Peru, dating back to 10,000 BP is argued by Dillehay et al. (2007).

References other than those already cited in Lenton et al.

Bell, E.A., Boehnke, P., Harrison, T.M., and Mao, W. L. 2015. Potentially biogenic carbon preserved in a 4.1 billion-year-old zircon. Proc. Natl. Acad. Sci. U.S.A. 112 (47): 14518-14521.

BP. 2015. BP Statistical Review of World Energy June, bp.com/statisticalreview.

Dillehay, T.D., Rossen, J., Andres, T.C. and Williams, D.E. 2007. Preceramic Adoption of Peanut, Squash, and Cotton in Northern Peru. Science 316: 1890-1893.

Kardashev, N. 1964. Transmission of Information by Extraterrestrial Civilizations. Soviet Astronomy 8: 217 – 221.

Malm, A. 2016. Fossil Capital. Verso: London.

Schwartzman, D. 1996. Solar Communism. Science & Society 60 (3): 307 - 331.

Schwartzman, D. 1999, 2002: Life, Temperature, and the Earth: The Self-Organizing Biosphere, Columbia University Press: New York.

Schwartzman, D. 2008. The limits to entropy: continuing misuse of thermodynamics in

environmental and Marxist theory. Science & Society 72 (1): 43 - 62.

Schwartzman, D. 2009a. Ecosocialism or Ecocatastrophe? Capitalism Nature Socialism 20 (1): 6 - 33.

Schwartzman, D. 2009b. Response to Naess and Hoyer. Capitalism Nature Socialism 20 (4): 93 - 97.

Schwartzman, D. 2012. A critique of degrowth and its politics. Capitalism Nature Socialism 23 (1): 119 - 125.

Schwartzman, D. 2014a. Revolutions that Made the Earth. Book Review. Am. J. Phys. 82: 529 - 530.

Schwartzman, D. 2014b, Is zero economic growth necessary to prevent climate catastrophe? Science & Society 78 (2): 235 -240.

Schwartzman, D. 2014c. My response to Trainer. Capitalism Nature Socialism 25 (4): 109 - 115.

Schwartzman, D. 2015a. The case for a hot Archean climate and its implications to the history of the biosphere. Posted April 1, arxiv.org.

Schwartzman, D. 2015b. The Geobiology of Weathering: a 13th Hypothesis. Posted September 14, arxiv.org.

Schwartzman, D. 2015c. From the Gaia hypothesis to a theory of the evolving self-organizing biosphere. Metascience 24 (2): 315-319.

Schwartzman, D. and Schwartzman, P. 2013. A rapid solar transition is not only possible, it is imperative!, African Journal of Science, Technology, Innovation and Development 5 (4): 297-302.

Schwartzman, P. and Schwartzman, D. 2011. A Solar Transition is Possible. Institute for Policy Research & Development Report, http://iprd.org.uk, solarUtopia.org.

Schwartzman D. and Volk, T. 1989. Biotic Enhancement of Weathering and the Habitability of Earth. Nature 340: 457-460.

---

## Referee Comment (RC2) · Anonymous Referee #2 · 2 Feb 2016

This is a well-argued and fascinating paper which I enjoyed reading very much. It presents a new approach to measuring the size of human society, its resource use and environmental impact against a natural background. The paper provides a long term perspective on energy transitions from the 3-4 Ga to present, brilliantly depicting six energetic revolutions, three in earth history and three in human history. The authors present a calculation of the resulting increase in energy input to the biosphere and human societies, respectively. This allows comparing the orders of magnitude of energy use by human society with energy inputs to the entire biosphere. Based on this, the paper establishes a link between energy flows and global material cycles and the associated environmental changes (e.g. by shifts in and scaling up of metabolic waste production) and the resulting feedbacks on energy capturing.

The paper is unique as it tries to link a long term earth history perspective with a short

term socioeconomic perspective on energy revolutions. In doing so, the paper provides a comprehensive overview of the current knowledge on energy revolutions in earth and human history, integrating knowledge from typically separate disciplinary approaches. What makes the paper innovative and outstanding is not so much new empirical evidence, but the attempt to link the different approaches and time scales of analysis from earth history and human environmental history. The paper impressively shows remarkable similarities between the six energy revolutions, each involving a new mechanism to capture free energy and accessing of previously underutilized sources. It shows how material constraints became limiting to the expansion of the scale of energy flows, either because of negative feedbacks of increasing waste outflows or because resources become scarce. From such a perspective, the ability to recycle the involved materials was crucial even in earth history revolutions. The analysis also underlines that the capacity of humans to push energy inputs towards planetary scales only emerged with industrial revolution. From the analysis of the past energy revolutions the authors draw lessons for a next energy revolution, the need for a closure of disrupted material cycles and the implications for socio-economic development.

Overall this is a well-structured and written, highly innovative and truly interdisciplinary paper - as far as I know, it is the first attempt to link earth and human history perspectives on energy transitions and their impact on the biosphere. The paper yields crucial insights for current debates about the Anthropocene, Planetary Boundaries and socio-economic metabolism research. I recommend to accept the paper for piublication and only have a few very minor remarks.

4/103 provide reference!

6/173 . . .lower weathering fluxes of phosphorus. . . (lower compared to what reference level? Why lower?

6/189 what is meant by "sinking export production" here?

10/330 looking at the information provided in Figure 2 I would say that energy use in

industrial is two orders of magnitude higher than the biological metabolic requirements of humans (300 GJ/cap/yr compared to ca. 3 GJ/ca/yr).

11/343 . . .accounted for as socioeconomic input. . .

11/363 – vulnerable to the scale and quantity of metabolic changes. . .. Unclear what this refers to.

12/405f –some references would be useful here

13/414 – something wrong with the sentence

Figure 2: Provide source information in Figure captions

16/525 – I think Krausmann et al. 2013 is the wrong reference here. This paper is about human appropriation of NPP!

16/529 – provide reference for CO2 emissions

16/540 – in a recent paper in the Journal of Industrial Ecology, Haas et al. (2015) have estimated that only 6% of all globally extracted materials are currently recycled within the socioeconomic system.

18/590 – Specify year of reference: Do the 25%-75% refer to current or projected (2050) energy demand.

18/580ff – The authors discuss the transition to a solar powered industrial energy system (biomass, PV, wind etc.). Even though I agree that solar powered technologies (PV, wind, biomass) are the way to go, I would appreciate a statement here on the possible significance and limitations of nuclear energy sources (fusion/fission) in future energy provision (based on the framework applied in the article) – These technologies are considered of high relevance in some sustainable energy strategies (see e.g. the recently published ecomodernist manifesto http://www.ecomodernism.org/ ) and should be addressed in one way or another!

---

## Referee Comment (RC3) · Anonymous Referee #3 · 9 Feb 2016

Reviewer comments on manuscript:

Revolutions in energy input and material cycling in Earth history and human history Timothy M. Lenton, Peter-Paul Pichler, and Helga Weisz

Overall this is a super-cool, excellent paper. I much enjoyed reading it twice, and have found it a rich source of detailed numbers, of the knowns and unknowns about the history of energy and material cycling over billions of years, and ideas about how we got to the current state of the planet and what might be needed for the future. I generally agree with their push for "scaling up new solar energy technologies and the development of much more efficient material recycling systems – thus creating a more autotrophic social metabolism" – a 7th revolution!

A few suggestions:
[Figure]

(1) Though "solar" is mentioned in the abstract, in the concluding section about renewables, they are apparently talking about wind, too, as a form of solar energy. They might want to be more clear about this.

(2) I found the dates in Table S1 to be confusing. The dates given for any item seem to be the final date for that "era" and the start of the next revolution. For example, the date of the Neolithic Revolution is given as 1850 AD (perhaps "CE" is more preferred here?) but really that is the start of the Industrial Revolution. I assume the dates refer to the time when their numbers apply, when the revolution is mature. I think this should be clarified. The same comments apply to the other revolutions, as dated in that table. (And the same comments apply to the year numbers in their Figure 1.)

(3) The authors might want to clarify their term "biosphere." They seem to apply it to what some might call the "biota," for example, the marine and terrestrial biota. Often the word biosphere is used to include all life, atmosphere, ocean, and soil, and thus the energy input to that would be the entire absorbed solar energy, which is not relevant to the authors' points and calculations.

(4) More on terminology: In Figure 1 the authors use "land plants" for the revolution or era they call "eukaryotic photosynthesis" in Table S1. These should be consistent, and given the discussion and calculation, "land plants" seems to be the better choice for what they are covering.

(5) Good discussion of how to draw the system boundary for human society on page 11.

(6) The authors might want to give more context for they think they have done, possibly at the end of the paper. For example, on page 2, they say, "following pioneering work by Smil (1991), we propose an alternative approach to measure the human influence against a natural background." It would be good to give some more credit to Smil: What was his pioneering approach? What have the authors taken from Smil, and what have they extended in their own work? This could be done with additional text on page 2, or,

again as I suggest, toward the end of the paper.

(6) All in all, wonderfully clear and interesting!

———————————————

---

## Author Comment (AC1) · 16 Mar 2016

**Authors' response to the reviewer comments by David Schwartzman, and 2 anonymous reviewers:**

*General response to all reviewers:*

*We thank all three reviewers for their enthusiastic, appreciative, detailed, and knowledgeable comments. We very much enjoyed reading through your comments and we learned a lot from them. We are aware that it is very ambitious to cover such broad and diverse scholarship in a single paper. In our responses to the reviewer's comments we aimed at keeping the big picture that motivated this paper in the first place, while adding more detail, addressing neglected aspects and being semantically more precise. Obviously we cannot fully cover all suggestions without jeopardizing the readability of the manuscript.*

*We hope this will be the beginning of a very exciting debate on important and fundamental aspects of the functional relations between energy and material use in the biosphere and in human societies.*

*Below are our point-to-point replies to the reviewer comments.*

*Legend:*

Reviewer's original comments in normal font
*Our replies in italic*
*"Our suggestions for ms revisions in italic and between quotation marks"*

REVIEWER 1 = David Schwartzman

I really enjoyed reviewing this paper both for its subject and for its comprehensive approach and in-depth scholarship, plus I got a chance to present my own views to those reading this online journal. This paper should be a valuable source for researchers, while the culminating section on a solar powered recycling revolution should be an inspiration to both scientists and activists around the world, a call to make this other world that is possible come into being, before our children and grandchildren are plunged into the abyss of climate catastrophe. Wind and solar power are potentially the energy source of a 21st century industrial revolution, a reprise of the role of coal in the 18th century apparently a result of capital's drive to control labor power versus using its alternative water mills (Malm, 2016).

*We thank David Schwartzman his generous and enthusiastic response to our paper.*

The paper's underlying theme is the succession of revolutions in Earth history, following Lenton and Watson's (2011) earlier invocation of this concept (see e.g., my review, Schwartzman, 2014a). The authors brilliantly illuminate each revolution, focusing on the stepwise increase in capture of energy by the biosphere (see their Figure 1). But not examined in this paper is whether these revolutions were essentially inevitable products of biospheric evolution, roughly deterministic or rather unique to Earth history. This question is of intense interest to the astrobiology community and its critics. For example, is the emergence of oxygenic photosynthesis very likely, given the likely abundance of water on Earth-like planets around Sun-like stars? I side with the determinists, play the tape again, pretty close to the same outcome (Schwartzman, 1999,2002; 2015c), a hypothesis likely to be tested in the next few decades by the exoplanetary research program. Further, in view of the order of magnitude billion year lag times in these revolutions, culminating in the very condensed outcomes in human history, were there constraints responsible for holding them

back for so long or was this timing simply a matter of chance as Lenton and Watson (2011) argue?

*The topic of determinism (or not) in the sequence of biospheric evolution is indeed a profound one, but we feel it is beyond the present paper to take a stance on this. Our aim instead is to document the energy revolutions.*

I have pointed out the constraint commonly ignored, the climatic temperature history of the geologic past. While still under debate, I find the oxygen isotope record in sedimentary chert and the compelling case for a near constant isotopic oxygen composition of seawater over geologic time supporting thermophilic surface temperatures prevailing in the Archean. A cold Archean is hard to explain given the likely higher outgassing rates of carbon dioxide, significantly smaller land areas and weaker biotic enhancement of weathering than present in the context of the long-term carbon cycle, taking into account the fainter Archean sun in climate modeling. This evidence points to an important conclusion regarding biological evolution, namely to the critical role of a temperature constraint holding back the emergence of major organismal groups, starting with phototrophs, culminating with metazoans in the latest Precambrian. As a result of the co-evolution of life and its abiotic environment, the evolution of Earth's biosphere is close to being deterministic, i.e., its origin and history and the general pattern of biotic evolution are very probable, given the same initial conditions, potentially a model for Earth-like planets around Sun-like stars (Schwartzman, 1999, 2002; 2015a).

*The interpretation of the oxygen isotope record of cherts remains contentious with relatively recent studies in Nature arguing that Archean ocean temperatures were <40 C from both oxygen and hydrogen isotopes in chert (Hren et al. 2009) or 26-35 C from oxygen isotopes in phosphate (Blake et al. 2010). We agree that outgassing rates were likely higher, land areas smaller, and the biotic enhancement of weathering much weaker in the Archean. However, our own modelling of the long-term carbon cycle, including seafloor weathering as a crucial alternative sink for $CO_2$ which is also temperature sensitive, agrees with these warm (but not hot) inferences of Archean conditions. In essence, with presumed higher seafloor spreading rates, seafloor weathering does much of the balancing of the $CO_2$ cycle. Thus, whilst we find the concept of high temperatures holding back biospheric evolution a very stimulating provocation, which might yet turn out to be correct, we have chosen not to discuss it within the paper.*

*http://www.nature.com/nature/journal/v462/n7270/full/nature08518.html*
*http://www.nature.com/nature/journal/v464/n7291/full/nature08952.html*

Briefly mentioned is the role of biotic enhancement of weathering (BEW) in triggering snow ball Earth episodes in the late Proterozoic (lines 274-275 citing Lenton and Watson, 2004). But evidence for the progressive increase in the BEW over geologic time has profound implications for the coevolution of the biosphere and its biota, since BEW represents a powerful catalytic factor in the long-term carbon cycle (Schwartzman and Volk, 1989; Schwartzman, 1999, 2002; 2015b). The Archean (and Hadean) was likely near abiotic with respect to BEW, so this factor must be taken into account in modeling the long-term carbon cycle in the context of abiotic drivers, namely rising solar luminosity, continental area and decreasing volcanic outgassing over geologic time to the present.

*We agree about the importance of the biotic enhancement of weathering. As noted above we do consider its changing strength alongside the other factors mentioned in our modelling of the long-term carbon cycle.*

Now for revolutions in human history.

Line 567. "The correlation between energy use and human development appears to be highly nonlinear.  At low levels of human development relatively small increases in energy input have large positive effects, while at high levels of human development large increase in energy input have little or no effect on further increases in standards of living (Steinberger and Roberts, 2010)."

Nevertheless, supplying the rough minimum of 3.5 kilowatt per person to the energy-deprived global South could dramatically increase life expectancies, arguably a robust measure of quality of life (Schwartzman and Schwartzman, 2011; 2013).

*We agree and this is quite consistent with what we have written. We have added a qualifier as follows "At high levels of human development large increases in energy input have little or no effect on further increases in standards of living. However, at low levels of human development relatively small increases in energy input have large positive effects (Steinberger and Roberts, 2010), for example supplying 3.5 kilowatt per person can greatly increase life expectancy (Schwartzman and Schwartzman, 2013)."*

Line 570. "Forward look: A solar powered recycling revolution" is a welcome end section to this paper.  I have long been advocating the same approach, recognizing that besides avoiding the well-know negative impacts of fossil fuels and and more contentiously nuclear power, high efficiency collection of solar radiation with wind and solar technologies has the capacity to do the work required for recycling.  The energy base of the global physical economy is critical: global wind/solar power will pay its "entropic debt" to space as non-incremental waste heat, unlike its unsustainable  alternatives (Schwartzman, 1996, 2008, 2009).

*We agree and have added the following sentence: "With regards to the lasting attention to Georgescu-Roegen's flawed fourth law of thermodynamics (Georgescu-Roegen, 1971) it is important to note that such a "revolution" does not contradict any established thermodynamic laws (Fleissner and Hofkirchner, 1994; Ayres, 1999; Schwartzman, 2008) as is amply demonstrated by the biological evolution of the Earth's biota."*

Line 580-590.
I find the authors being too conservative in their assessment given the challenges humanity is facing in this century. Going beyond business-as-usual modeling being cited here is imperative. Radical changes in global political economy are likely necessary to implement a timely and robust energy transition to wind/solar power coupled with the elimination of fossil fuels. In thirty years or less this transition could deliver two times the global primary energy consumption level of 551 EJ yr−1 in 2014 (BP, 2015), equivalent to 35 Tera Watts, even with present efficiencies in collection which will increase in this time frame (Schwartzman and Schwartzman, 2011; Schwartzman, 2014). Assuming modest population growth by 2050 corresponding to roughly 9 billion, this level will be needed to terminate energy poverty in the global South, insuring state-of-the-science life expectancy for all of humanity, as well as to generate the incremental energy required for carbon-sequestration from the atmosphere and climate adaptation..

We are now a Type I civilization in Kardashev's scale of cosmic civilizations (Kardeshev, 1964), at a bifurcation, an imminent choice between the collapse of civilization or the emergence of a truly planetary civilization mobilizing our star's fusion energy for human and nature's needs instead of our present reality of perpetual war on both entities. I submit that an approximate doubling of global energy consumption using this energy source is a necessary condition for the better choice.

*We concede that we may have been conservative in our assessment of renewable energy potential to 2050, relying as we did on published energy-economy integrated assessment model studies. To balance things out we have added a paragraph on the dependence of those studies on the*

*underlying assumptions, especially on the assumptions about future energy demand and technological development (see below our reply to the last comment from reviewer 2).*

Lines 641-644. "An outstanding task therefore is to formulate a steady-state "Earth system economics" that supports long-term human and planetary well-being. One of the most difficult problems to be solved along the way will be to find out how a steady-state society can find new ways to organize a just distribution of wealth in an economy that functions physically as a zero sum game."

Commonly "steady-state" is taken as referring to a no-growth economy. However, the qualitative versus quantitative aspects of economic growth should be distinguished, with the concept of economic growth being deconstructed, particularly with respect to ecological and health impacts. Growth of what are we speaking, weapons of mass destruction, unnecessary commodities, SUVs versus bicycles, culture, information, pollution? Instead, advocates of global degrowth with their goal of reaching a zero growth steady-state economy commonly lump all growth into a homogenous outcome of the physical and political economy (Schwartzman, 2009, 2012). A zero growth economy is a unwelcome prescription for the immediate challenges posed by the threat of catastrophic climate change as well the undeniable lack of material consumption enjoyed by the majority of humanity living in the global South, the lack of adequate nutrition, housing, education and provision for health services, but most critically, their state of energy poverty. A sustainable growth phase, beginning in capitalism itself, must necessarily have a different quality than capitalist economic growth as measured by the GNP, namely not only requiring global growth in the wind and solar power infrastructure, but also in the agroecological sector. Sustainable economic growth would include global solarization of energy supplies, demilitarization and ecosystem repair (Schwartzman 2009).

*We are sympathetic to David's arguments here. We are not advocating an interval of degrowth, rather we are trying to anticipate here a stable and desirable end state for current growth and development. We recognize that in a steady-state global economy although physical components are constrained, non-physical aspects such as knowledge can continue to grow. We are also aware of the problems of measuring growth and wellbeing with GDP and of the undesirable consequence a GDP focused degrowth policy would have on the world's poorest population (which amounts to 1-2 billion people, depending on the definition of extreme poverty). While we cannot go deeply into this debate in the present paper, we have changed the last sentence to better explain our position: "An outstanding task therefore is to formulate a steady-state "Earth system economics" that supports long-term human and planetary well-being. Two of the most difficult problems to be solved along the way will be to find out how desirable attributes of society, such as knowledge, can still grow while material input is constrained and how to organize a just distribution of access to material and non-material resources in an economy that functions as a physical zero sum game."*

More specific comments

Line 89. Update: evidence for biogenic carbon in 4.1 Ga zircons (Bell et al., 2015).

*Thanks – we have added "and perhaps as early as 4.1 Ga (Bell et al. 2015)" noting that the authors themselves are somewhat circumspect about whether they have really found biogenic carbon.*

Line 131-139. The authors are correct to point out the ambiguity of the sedimentary isotopic record of carbon; also see my discussion in Schwartzman (1999, 2002), p. 26-31. A higher volcanic outgassing rate of carbon in the Archean was balanced by a more intense silicate weathering sink on smaller continents and oceanic islands, plausibly releasing a higher flux of nutrients to the ocean (Schwartzman, 1999, 2002; 2015b). Hence marine biotic productivity may have been

enhanced, and biospheric energy capture in the Precambrian may be underestimated.

*We have added citation to Schwartzman (1999) regarding the interpretation of the carbon isotope record. Regarding the weathering argument, our own modelling suggests that sea-floor weathering was relatively speaking a more important carbon sink than continental silicate weathering in the Archean (Mills et al. 2014, and see also Sleep and Zahnle 2001). Current understanding is that seafloor weathering is not a source of phosphorus, in fact phosphorus is consumed at mid-ocean ridges where basalt is emplaced. Therefore we think nutrient limitation was probably more acute in the early Precambrian. However the point we are trying to make here is that an anoxygenic photosynthetic biosphere is unlikely to have been nutrient limited, rather it would be limited by the supply of electron donors for photosynthesis.*

Line 276. Modern desert microbial crust productivity is not a good model for Proterozoic or early Phanerozoic terrestrial biota.

*This is a fair point given low productivity in desert environments. We also cited a relatively low estimate of global NPP from a model of cryptogamic cover. However, recent work that T.M.L. is involved in applying the same ecophysiological model to a high $CO_2$ world without vascular plants has shown that a world of cryptogamic cover could have approached ~25% of today's global NPP. We have reworked this part accordingly: "Estimates of the productivity of global microbial mats, based on a simple area-scaling of modern desert crust (Brostoff et al., 2005), suggests only 3–11% of today's terrestrial NPP, comparable to today's cryptogamic cover, which achieves 1–6% of terrestrial NPP (Porada et al., 2013). However, deserts are unproductive environments and modern cryptogamic cover is living in a world dominated by vascular plants. Taking the ecophysiological model of cryptogamic cover (Porada et al., 2013) and considering higher atmospheric $CO_2$ and lack of competition from vascular plants, putative Neoproterozoic-early Paleozoic land biota might have achieved ~25% of today's terrestrial NPP."*

Line 416. Even earlier agriculture in Peru, dating back to 10,000 BP is argued by Dillehay et al. (2007).

*Thanks for alerting us to this paper we now cite it and have changed our account of the dates and places of the origin of agriculture into: "Agriculture had multiple independent origins; in the Near East (~10000 BP), Peru (~10000 BP), South China (8500 BP), North China (7800 BP), Mexico (4800 BP), East North America (4500 BP), and possibly sub-Saharan Africa (4000 BP) (Smith, 1995; Dillehay et al., 2007; Diamond and Bellwood, 2003; Barker, 2006)." We have also added a reference to a recent paper on evidence for earlier agriculture in the Persian Gulf: "Archaeological evidence from several sites at the shoreline of the Persian Gulf has revealed a rapid colonization of this area by advanced agricultural and urban societies at around 7500 BP. As sea-level rise from the last glacial low stand was only completed in the Persian Gulf 7000-8000 BP, there could be even older agricultural sites in areas that are now beneath the Indian Ocean (Rose, 2010)."*

References other than those already cited in Lenton et al.

Bell, E.A., Boehnke, P., Harrison, T.M., and Mao, W. L. 2015. Potentially biogenic carbon preserved in a 4.1 billion-year-old zircon. Proc. Natl. Acad. Sci. U.S.A. 112 (47): 14518-14521.

BP. 2015. BP Statistical Review of World Energy June, bp.com/statisticalreview.

Dillehay, T.D., Rossen, J., Andres, T.C. and Williams, D.E. 2007. Preceramic Adoption of Peanut, Squash, and Cotton in Northern Peru. Science 316: 1890-1893.

Kardashev, N. 1964.  Transmission of Information by Extraterrestrial Civilizations.  Soviet Astronomy 8: 217 – 221.

Malm, A. 2016. Fossil Capital. Verso: London.

Schwartzman, D. 1996. Solar Communism. Science & Society 60 (3): 307 - 331.

Schwartzman, D. 1999, 2002:  Life, Temperature, and the Earth:  The Self-Organizing Biosphere, Columbia University Press: New York.

Schwartzman, D. 2008. The limits to entropy: continuing misuse of thermodynamics in environmental and Marxist theory. Science & Society 72 (1): 43 - 62.

Schwartzman, D. 2009a. Ecosocialism or Ecocatastrophe? Capitalism Nature Socialism 20 (1): 6 - 33.

Schwartzman, D. 2009b. Response to Naess and Hoyer. Capitalism Nature Socialism 20 (4): 93 - 97.

Schwartzman, D. 2012.  A critique of degrowth and its politics.  Capitalism Nature Socialism 23 (1): 119 - 125.

Schwartzman, D. 2014a. Revolutions that Made the Earth. Book Review. Am. J. Phys. 82: 529 - 530.

Schwartzman, D. 2014b, Is zero economic growth necessary to prevent climate catastrophe? Science & Society 78 (2): 235 -240.

Schwartzman, D. 2014c. My response to Trainer. Capitalism Nature Socialism 25 (4):109 - 115.

Schwartzman, D. 2015a. The case for a hot Archean climate and its implications to the history of the biosphere. Posted April 1, arxiv.org.

Schwartzman, D. 2015b.  The Geobiology of Weathering:  a 13th Hypothesis.  Posted September 14, arxiv.org.

Schwartzman, D. 2015c.  From the Gaia hypothesis to a theory of the evolving self-organizing biosphere. Metascience 24 (2): 315-319.

Schwartzman, D. and Schwartzman, P. 2013. A rapid solar transition is not only possible, it is imperative!, African Journal of Science, Technology, Innovation and Development 5 (4): 297-302.

Schwartzman, P. and Schwartzman, D. 2011.  A Solar Transition is Possible.  Institute for Policy Research & Development Report, http://iprd.org.uk, solarUtopia.org.

Schwartzman D. and Volk, T. 1989. Biotic Enhancement of Weathering and the Habitability of Earth. Nature 340: 457-460.

REVIEWER 2

This is a well-argued and fascinating paper which I enjoyed reading very much. It presents a new approach to measuring the size of human society, its resource use and environmental impact against a natural background. The paper provides a long term perspective on energy transitions from the 3-4 Ga to present, brilliantly depicting six energetic revolutions, three in earth history and three in human history. The authors present a calculation of the resulting increase in energy input to the biosphere and human societies, respectively. This allows comparing the orders of magnitude of energy use by human society with energy inputs to the entire biosphere. Based on this, the paper establishes a link between energy flows and global material cycles and the associated environmental changes (e.g. by shifts in and scaling up of metabolic waste production) and the resulting feedbacks on energy capturing.

The paper is unique as it tries to link a long term earth history perspective with a short term socioeconomic perspective on energy revolutions. In doing so, the paper provides a comprehensive overview of the current knowledge on energy revolutions in earth and human history, integrating knowledge from typically separate disciplinary approaches. What makes the paper innovative and outstanding is not so much new empirical evidence, but the attempt to link the different approaches and time scales of analysis from earth history and human environmental history. The paper impressively shows remarkable similarities between the six energy revolutions, each involving a new mechanism to capture free energy and accessing of previously underutilized sources. It shows how material constraints became limiting to the expansion of the scale of energy flows, either because of negative feedbacks of increasing waste outflows or because resources become scarce. From such a perspective, the ability to recycle the involved materials was crucial even in earth history revolutions. The analysis also underlines that the capacity of humans to push energy inputs towards planetary scales only emerged with industrial revolution. From the analysis of the past energy revolutions the authors draw lessons for a next energy revolution, the need for a closure of disrupted material cycles and the implications for socio-economic development.

Overall this is a well-structured and written, highly innovative and truly interdisciplinary paper - as far as I know, it is the first attempt to link earth and human history perspectives on energy transitions and their impact on the biosphere. The paper yields crucial insights for current debates about the Anthropocene, Planetary Boundaries and socio-economic metabolism research. I recommend to accept the paper for publication and only have a few very minor remarks.

*We thank the reviewer for the generous and enthusiastic response.*

4/103 provide reference!

*Thanks – we have added reference to Canfield et al. (2006).*

6/173 lower weathering fluxes of phosphorus (lower compared to what reference level? Why lower?

*Lower than present, because of a shift in the relative importance of terrestrial and seafloor weathering in balancing the carbon cycle – our model predicts a greater role for seafloor weathering in the past, consistent with some other studies (e.g. Sleep and Zahnle, 2001), leading to a corresponding reduction in phosphorus supply (because seafloor weathering is not a source of P). We have altered the corresponding text as follows: "Lower terrestrial weathering fluxes of phosphorus (relative to present) have been predicted, due to a shift from terrestrial to seafloor weathering to balance the carbon cycle earlier in Earth history, and this would have tended to reduce ocean phosphorus concentration, because seafloor weathering is not a source of phosphorus (Mills et al., 2014)."*

6/189 what is meant by "sinking export production" here?

*Apologies – this is oceanographer's terminology for the flux of carbon that sinks out of the well-mixed surface layer of the ocean. We have rephrased as follows: "A large flux of methane, equivalent to around 60% of the primary production sinking out of the surface layer of the ocean (Daines and Lenton, 2016)."*

10/330 looking at the information provided in Figure 2 I would say that energy use in industrial is two orders of magnitude higher than the biological metabolic requirements of humans (300 GJ/cap/yr compared to ca. 3 GJ/ca/yr).

*Oh yes, stupid mistake! Thanks for your careful reading. Corrected.*

11/343 accounted for as socioeconomic input

*Well, we thought this was already clear from the context. Still we added the suggested qualifier as it improves clarity.*

11/363 – vulnerable to the scale and quantity of metabolic changes -  Unclear what this refers to.

*Apologies, we have not elaborated this argument in sufficient clarity. We have rephrased this sentence which now reads: "The emergence and continued existence of human civilization is conditional upon the stability of certain basic dynamics of the Earth system which are vulnerable to metabolic changes in kind and scale, such as changes in the overall energy balance, or changes in the chemical composition of the atmosphere, oceans or soils, rather than the specific mechanisms that caused them."*

12/405 –some references would be useful here:

*We have added references to (Headland et al, 1989, Gowdy 1997):*

*Headland TN, et al. (1989) Hunter-Gatherers and Their Neighbors from Prehistory to the Present. Current Anthropology 30(1):43–66.*

*Gowdy J (1997) Limited Wants, Unlimited Means: A Reader On Hunter-Gatherer Economics And The Environment. Island Press.*

13/414 – something wrong with the sentence

*We have rephrased this sentence to: "Agriculturalists greatly enhanced the area productivity of edible species at the expense of non-edible species and of food competitors. In contrast to pre-agricultural societies they lived in a controlled solar energy system (Sieferle, 1997)."*

Figure 2: Provide source information in Figure captions

*Corrected.*

16/525 – I think Krausmann et al.  2013 is the wrong reference here.  This paper is about human appropriation of NPP!

*Thanks for catching this mistake. We now correctly refer to (Krausmann et al. 2013B, 2009)*

*Krausmann, F., Gingrich, S., Eisenmenger, N., Erb, K.-H., Haberl, H., and Fischer-Kowalski, M.: Growth in global materials use, GDP and population during the 20th century, Ecological Economics, 68, 2696–2705, doi:10.1016/j.ecolecon.2009.05.007, http://www.sciencedirect.com/science/article/pii/S0921800909002158, 2009.*

*and*

*Krausmann, F., Schaffartzik, A., Mayer, A., Gingrich, S., and Eisenmenger, N.: Global trends and patterns in material use, in: Symposium K – Materials for Sustainable Development, vol. 1545 of MRS Online Proceedings Library, doi:10.1557/opl.2013.1075, http://journals.cambridge.org/article_S1946427413010750, 2013b.*

16/529 – provide reference for CO2 emissions

*We now provide the reference:*

*Marland, G., Boden, T. A., Andres, R. J., Brenkert, A. L., and Johnston, C. A.: Global, regional, and national fossil fuel CO2 emissions, Trends: A Compendium of Data on Global Change, pp. 37 831–6335, http://cdiac.ornl.gov/trends/emis/overview, 2007.*

16/540 – in a recent paper in the Journal of Industrial Ecology, Haas et al. (2015) have estimated that only 6% of all globally extracted materials are currently recycled within the socioeconomic system.

*We have added this reference.*

18/590 – Specify year of reference: Do the 25%-75% refer to current or projected (2050) energy demand.

*Added "projected (2050)" to the sentence.*

18/580ff – The authors discuss the transition to a solar powered industrial energy system (biomass, PV, wind etc.). Even though I agree that solar powered technologies (PV, wind, biomass) are the way to go, I would appreciate a statement here on the possible significance and limitations of nuclear energy sources (fusion/fission) in future energy provision (based on the framework applied in the article) – These technologies are considered of high relevance in some sustainable energy strategies (see e.g. the recently published ecomodernist manifesto http://www.ecomodernism.org/ ) and should be addressed in one way or another!

*This is a good point, and we acknowledge that nuclear energy sources could play an important and complementary role to solar powered technologies in future energy provision (e.g. providing baseload power to electricity grids). We made the following addition to the ms: "The importance of other, more contested energy technologies for achieving a sustainability transition of the global energy system depends on the development of future energy demand. Assuming ambitious energy efficiency improvements the transformation goals can be achieved without nuclear fission, carbon-capture and storage, or high-tech carbon sink management. With less progress on the demand side, one or more of these technologies would be required in the energy mix (Riahi et al., 2012). Nuclear fusion might be an option in the long-term, but is not an attainable option in the coming decades when climate mitigation measures must be implemented (World Bank, 2012). Significant additional investments and several decades of technology development would be needed to bring nuclear fusion into large scale practical implementation (von Hippel et al., 2012)."*

*Riahi K, et al. (2012) Chapter 17 - Energy Pathways for Sustainable Development. Global Energy Assessment - Toward a Sustainable Future (Cambridge University Press, Cambridge, UK and New York, NY, USA and the International Institute for Applied Systems Analysis, Laxenburg, Austria), pp 1203–1306.*

*von Hippel F, et al. (2012) Chapter 14 - Nuclear Energy. Global Energy Assessment - Toward a Sustainable Future (Cambridge University Press, Cambridge, UK and New York, NY, USA and the International Institute for Applied Systems Analysis, Laxenburg, Austria), pp 1069–1130.*

*World Bank (2012) Turn down the heat : why a 4°C warmer world must be avoided (The World Bank).*

REVIEWER 3

Reviewer comments on manuscript:

Overall this is a super-cool, excellent paper. I much enjoyed reading it twice, and have found it a rich source of detailed numbers, of the knowns and unknowns about the history of energy and material cycling over billions of years, and ideas about how we got to the current state of the planet and what might be needed for the future. I generally agree with their push for "scaling up new solar energy technologies and the development of much more efficient material recycling systems – thus creating a more autotrophic social metabolism" – a 7th revolution!

*We thank the reviewer for their enthusiastic and positive response.*

A few suggestions:

(1) Though "solar" is mentioned in the abstract, in the concluding section about renewables, they are apparently talking about wind, too, as a form of solar energy. They might want to be more clear about this.

*We are indeed including all forms of solar energy including wind and biomass. We have clarified this and now refer to "renewable and decarbonized" energy technologies in the abstract and clarifying that in this context most renewable energy technologies are ultimately based on solar energy in section "Forward look: A solar powered recycling revolution".*

(2) I found the dates in Table S1 to be confusing. The dates given for any item seem to be the final date for that "era" and the start of the next revolution. For example, the date of the Neolithic Revolution is given as 1850 AD (perhaps "CE" is more preferred here?) but really that is the start of the Industrial Revolution. I assume the dates refer to the time when their numbers apply, when the revolution is mature. I think this should be clarified. The same comments apply to the other revolutions, as dated in that table. (And the same comments apply to the year numbers in their Figure 1.)

*We completely agree with the reviewer and have clarified this both in Fig. 1 and S1.*

(3) The authors might want to clarify their term "biosphere." They seem to apply it to what some might call the "biota," for example, the marine and terrestrial biota. Often the word biosphere is used to include all life, atmosphere, ocean, and soil, and thus the energy input to that would be the entire absorbed solar energy, which is not relevant to the authors' points and calculations.

*This is an important semantic point. We like the term 'biosphere' because of the global scale to life that it conveys, and in geochemistry 'biosphere' is often taken to be synonymous with 'biota' i.e. the sum total of all life. However, we recognize that if readers took the term to include all solar energy input it would be misleading. To clarify this we have added a definition to the introduction: "'biosphere' is taken here to be synonymous with the biota i.e. the sum total of all life on the planet"*

(4) More on terminology: In Figure 1 the authors use "land plants" for the revolution or era they call "eukaryotic photosynthesis" in Table S1. These should be consistent, and given the discussion and calculation, "land plants" seems to be the better choice for what they are covering.

*We agree and have change to 'land plants' in Tabl*e S1.

(5) Good discussion of how to draw the system boundary for human society on page 11.

*Thanks.*

(6) The authors might want to give more context for they think they have done, possibly at the end of the paper. For example, on page 2, they say, "following pioneering work by Smil (1991), we propose an alternative approach to measure the human influence against a natural background." It would be good to give some more credit to Smil: What was his pioneering approach? What have the authors taken from Smil, and what have they extended in their own work? This could be done with additional text on page 2, or, again as I suggest, toward the end of the paper.

*We do indeed owe a great debt to Smil. We have expanded the introduction to say more: "Here we propose an alternative approach to measure the human influence against a natural background, following pioneering work by Smil (1991), who first compared energy use in the biosphere and in human civilization (where 'biosphere' is taken here to be synonymous with the biota i.e. the sum total of all life on the planet). Our starting point is the fundamental ability of all life forms, from archaea and bacteria to human societies, to capture free energy and to use it for moving and transforming matter in order to sustain an internal order. Building on Smil's (1991, 2008) characterization of energy use in the biosphere and human civilization, we expand the temporal dimension to consider the full history of transitions in biospheric energy capture, and we add a material cycling dimension (also partly inspired by Smil's work). In both Earth and human history major revolutions in energy capture have occurred, with each subsequent transition resulting in higher energy input, altered material cycles and major consequences for the internal organization of the respective systems."*

(6) All in all, wonderfully clear and interesting!

*Thanks!*